# THINK SMART, NOT HARD: DIFFICULTY ADAPTIVE REASONING FOR LARGE AUDIO LANGUAGE MODELS

## ABSTRACT

Large Audio Language Models (LALMs), powered by the chain-of-thought (CoT) paradigm, have shown remarkable reasoning capabilities. Intuitively, different problems often require varying depths of reasoning. While some methods can determine whether to reason for a given problem, they typically lack a fine-grained mechanism to modulate how much to reason. This often results in a "one-size-fits-all" reasoning depth, which generates redundant overthinking for simple questions while failing to allocate sufficient thought to complex ones. In this paper, we conduct an in-depth analysis of LALMs and find that an effective and efficient LALM should reason smartly by adapting its reasoning depth to the problem's complexity. To achieve this, we propose a difficulty-adaptive reasoning method for LALMs. Specifically, we propose a reward function that dynamically links reasoning length to the model's perceived problem difficulty. This reward encourages shorter, concise reasoning for easy tasks and more elaborate, in-depth reasoning for complex ones. Extensive experiments demonstrate that our method is both effective and efficient, simultaneously improving task performance and significantly reducing the average reasoning length. Further analysis on reasoning structure paradigm offers valuable insights for future work.

## 1 INTRODUCTION

In recent years, general artificial intelligence advances rapidly with the development of large language models (LLMs) (Dubey et al., 2024; Hurst et al., 2024; Team et al., 2024; Team, 2024). The reasoning ability of LLMs is further enhanced by the chain-of-thought (CoT) paradigm (Jaech et al., 2024; DeepSeek-AI et al., 2025; Yang et al., 2025), which significantly improves performance on complex problems. At the same time, Large Audio Language Models (LALMs) (Tang et al., 2024; Chu et al., 2023; 2024) also progress rapidly and raise an important question: how can they achieve reasoning more efficiently and effectively?

In previous work, Xie et al. (2025) enable reasoning in LALMs through supervised fine-tuning (SFT) on a large-scale dataset with CoT annotations, providing the first evidence that CoT is effective for solving complex audio understanding problems. Building on this, Li et al. (2025) introduces group relative policy optimization (GRPO) into LALMs for the first time, surpassing the former with less data and offering initial comparisons across different prompts. These studies focus either on SFT or GRPO, and both apply reasoning to all questions, but they lack in-depth analysis of the differences between the two approaches. Therefore, we conduct a detailed analysis to determine under what conditions SFT and GRPO are more effective. We find that GRPO performs better on harder questions, whereas on easier ones it tends to produce redundant reasoning and performs slightly worse than SFT. Furthermore, we analyze GRPO under forced reasoning (explicit prompt) and without forced reasoning (implicit prompt), and observe that the forced reasoning models maintain a clear advantage on harder questions, while the two settings perform similarly on easier ones. Taken together, these findings indicate that achieving efficient reasoning in LALMs requires adapting reasoning length to problem difficulty.

Regarding reasoning length and efficiency, Qu et al. (2025) highlight issues such as redundancy and overthinking. RL-based studies on LLMs (Arora & Zanette, 2025; Aggarwal & Welleck, 2025) design length-penalty rewards but rely on fixed thresholds that overlook question types and difficulty levels. In the audio domain, Wu et al. (2025) introduces a "when to think" mechanism that guides the

model on whether reasoning is necessary. However, for the samples that still perform reasoning, it lacks a fine-grained mechanism to modulate how much to reason.Therefore, an effective and efficient LALM should reason smartly by adapting its reasoning depth to the problem's complexity, achieving short reasoning for simple questions and deeper reasoning for difficult ones. Based on this analysis and prior work, we propose a new length-based reward function that no longer depends on fixed thresholds. In addition, we introduce two difficulty-adaptive standards to complement this reward, enabling reasoning length to align more appropriately with question difficulty. Both approaches achieve strong performance on the MMAU benchmark (Sakshi et al., 2025), particularly on harder questions, while also producing much shorter reasoning than direct GRPO models, thereby greatly improving reasoning efficiency. Furthermore, we conduct a qualitative case study on models from the main experiments, providing a detailed analysis of the reasoning structures in their outputs.

In conclusion, the main contributions are as follows:

- In this paper, we conduct in-depth analyses of LALMs and show that a smart LALM should reason adaptively, adjusting its reasoning depth to match the complexity of the problem.
- Based on this analysis, we propose two difficulty-adaptive length-based rewards, aiming to shorten reasoning length while achieving concise reasoning for simple questions and encouraging deeper reasoning for difficult ones.
- We carry out extensive experiments on the two proposed rewards, both of which achieve strong performance on the MMAU benchmark while significantly reducing reasoning length. In addition, we provide a qualitative analysis of output paradigms across models, offering useful guidance for future work.

We have provided an anonymous GitHub repository `https://anonymous.4open.science/r/ICLR2026-Anonymous-Repo-24196/` to support the reproduction of this work and will release all our experimental models after the double-blind review process.

## 2 DEEP ANALYSIS OF DIFFERENT METHODS FOR LALMS

LALMs have already demonstrated strong capability in addressing basic understanding tasks. Recent studies (Xie et al., 2025; Li et al., 2025) focus on enhancing their ability to solve complex problems through reasoning. However, while these works adopt different implementation methods, they lack in-depth analysis of the differences between approaches. Therefore, in this section, we conduct detailed analysis from two perspectives: which is more effective under various conditions between SFT and GRPO, and whether performance gains are driven by explicit or implicit prompt.

**Data.** For training, we use two datasets, FS and AVQA. FS is constructed on the basis of AVQA by incorporating four additional datasets, covering three different task types. Its size is approximately twice that of AVQA, and the detailed distribution is presented in Table 1.

Table 1: The data distribution of the FS training set, including different task types and their corresponding data sources.

| Task | Dataset-Source | Num |
|---|---|---|
| Audio Grounding | AudioGrounding (Xu et al., 2021) | 1,805 |
| Sound Classification | VocalSound (Gong et al., 2022) | 15,531 |
| | TUT2017 (Mesaros et al., 2017) | 3,744 |
| Sound Question Answering | Clotho-AQA (Lipping et al., 2022) | 6,615 |
| | AVQA (Yang et al., 2022) | 36,036 |

**Setup.** For the base models, we primarily adopt Qwen2-Audio-7B-Instruct [1] and Qwen2.5-Omni-7B [2]. More detailed experimental settings, including the two prompts, are provided in Appendix

---

[1] https://huggingface.co/Qwen/Qwen2-Audio-7B-Instruct

[2] https://huggingface.co/Qwen/Qwen2.5-Omni-7B

A.2. At the same time, for those experiments that share common settings, we also add an additional group average in Table 2 to facilitate comparisons across groups.

## 2.1 How Does GRPO Reasoning Differ from Direct Answers with SFT?

Here we mainly compare the "SFT on Qwen2-Audio-7B-Instruct" part and the "GRPO on Qwen2-Audio-7B-Instruct" part of Table 2. From the "Average" of the SFT part and the "Average" of the Prompt2 part, we observe that SFT performs very well on easy-level questions, while GRPO shows greater advantages on medium and hard questions. We believe this is mainly because, when facing medium and harder questions, the model cannot rely only on the base knowledge learned during pretraining to give direct answers and instead needs to learn how to use this knowledge through reasoning. In contrast, its weaker performance on easy questions is largely due to redundant reasoning content or errors made during the reasoning process that are carried over into the final answer, leading to incorrect results.

Therefore, for the question in this subsection, we conclude that GRPO is more effective on complex tasks that models cannot solve directly, but its reasoning on simple ones still needs further optimization to reduce redundancy and potential error propagation.

Table 2: The performance of different models under different training paradigms, fine-tuning strategies, training datasets, and prompting styles.

| Models | Airbench-Foundation | MMAU-Test-Mini | | | | | | |
|---|---|---|---|---|---|---|---|---|
| | Sound | Sound | Music | Speech | Easy | Medium | Hard | Avg |
| SFT On Qwen2-Audio-7B-Instruct | | | | | | | | |
| On FS, Full | 80.49 | 59.76 | 58.38 | 62.46 | 57.36 | 64.31 | 54.31 | 60.20 |
| On AVQA, Full | 67.06 | 60.36 | 56.59 | 59.46 | 53.10 | 63.92 | 53.88 | 58.80 |
| On FS, LoRA | 77.74 | 68.17 | 65.57 | 64.86 | 59.30 | 74.51 | 55.60 | 66.20 |
| On AVQA, LoRA | 67.38 | 66.07 | 60.48 | 54.65 | 49.22 | 69.80 | 52.16 | 60.40 |
| Average | 73.16 | 63.59 | 60.25 | 60.35 | 54.74 | 68.13 | 53.96 | 61.40 |
| GRPO On Qwen2-Audio-7B-Instruct | | | | | | | | |
| On FS, Full, Prompt2 | 81.10 | 67.57 | 64.67 | 62.76 | 55.04 | 73.73 | 56.90 | 65.00 |
| On AVQA, Full, Prompt2 | 70.35 | 67.87 | 66.77 | 60.96 | 52.33 | 75.10 | 57.76 | 65.20 |
| On FS, LoRA, Prompt2 | 69.60 | 69.37 | 60.48 | 55.26 | 49.61 | 71.76 | 53.02 | 61.70 |
| On AVQA, LoRA, Prompt2 | 69.38 | 66.97 | 59.28 | 56.16 | 47.29 | 70.78 | 53.88 | 60.80 |
| Average | 72.61 | 67.95 | 62.80 | 58.79 | 51.07 | 72.84 | 55.39 | 63.18 |
| On FS, LoRA, Prompt1 | 69.87 | 67.27 | 61.98 | 56.76 | 50.00 | 71.57 | 54.31 | 62.00 |
| On AVQA, LoRA, Prompt1 | 68.10 | 64.86 | 60.18 | 53.15 | 47.29 | 71.18 | 50.43 | 60.20 |
| Average | 68.99 | 66.07 | 61.08 | 54.96 | 48.65 | 71.38 | 52.37 | 61.10 |
| GRPO On Qwen2.5-Omni-7B | | | | | | | | |
| On FS, Full, Prompt2 | **83.46** | 72.37 | **67.66** | 68.76 | 59.30 | **78.43** | **61.63** | **69.60** |
| On FS, LoRA, Prompt2 | 76.86 | **73.57** | 65.56 | **69.06** | **59.69** | 78.23 | 60.77 | 69.40 |
| Average | 80.16 | 72.97 | 66.61 | 68.91 | 59.49 | 78.33 | 61.20 | 69.50 |

## 2.2 Does Performance Come from Explicit Reasoning or Implicit Activation?

This part mainly compares the four LoRA experiments under the "GRPO on Qwen2-Audio-7B-Instruct" part. First, it should be noted that the two prompts produce outputs with clear differences. Models trained with the implicit prompt do not generate a reasoning process for nearly every sample as those trained with the explicit prompt do; instead, they often directly produce answers in a way similar to SFT. When comparing results within the "Prompt1" (implicit prompt) experiments, models trained on the larger FS dataset consistently achieve better performance. These results suggest that using implicit prompts introduces SFT-like characteristics, relying on larger datasets to achieve stronger generalization, whereas explicit reasoning allows the model to truly learn from the data, which in turn demonstrates the necessity of explicit reasoning. When further comparing "Prompt1"

and "Prompt2" (explicit prompt), the performance gap is generally small, with "Prompt2" outperforming "Prompt1" by about 0.15 on average. Looking at the details, "Prompt1" performs about 0.1 better on easy and medium questions, but lags behind by 1.1 on hard questions. These results further show that reasoning on easier questions can lead to redundancy and error propagation, while harder questions require deeper reasoning.

Overall, for the question in this subsection, we conclude that reasoning for all questions or not reasoning at all is not the optimal solution. Instead, the model should learn to adjust reasoning length according to different questions—reducing reasoning length for those that do not require it, while increasing reasoning for those that lack it. In this way, it can achieve both performance improvements and efficiency gains.

## 3 Enhancing LALMs with Difficulty-Aware Adaptive Reasoning

The above results and analysis indicate that different types of questions require different reasoning lengths. Therefore, we aim to link question difficulty with reasoning length, enabling shorter reasoning for simple questions and deeper reasoning for difficult ones. Specifically, we first define two model-perspective difficulty-adaptive standards: one based on group accuracy of rollout samples and the other based on the audio attention of the current sample. We then apply these difficulty standards to a rule-based reward function that varies with reasoning length, thereby linking question difficulty with reasoning length. In the following, we elaborate on these two components in detail.

### 3.1 Defining Model-perspective Difficulty

As mentioned above, our core idea is to encourage models to reason more on difficult questions and less on simple ones. Thus, our main approach is to dynamically adjust the reward based on both question difficulty and reasoning length. At the same time, since there are gaps between different definitions of difficulty perspectives, the model's own perspective during training can better reflect the actual situation. Therefore, in this subsection, we provide a detailed explanation of the two proposed model-perspective difficulty-adaptive standards. We refer to these as **G**roup **R**atio **D**ifficulty **R**eward (GRDR) and **G**roup **A**udio **A**ttention **D**ifficulty **R**eward (GA$^2$DR).

**GRDR.** The first is based on the ratio of correct samples within a rollout group. For example, when the group size $G = 8$, if six or more responses are correct, the question is labeled as easy; if fewer than six but at least three are correct, it is labeled as medium; and if fewer than three are correct, it is labeled as hard. $\gamma$ is used to represent the difficulty value of a question, where larger values correspond to more difficult questions. The corresponding $\gamma$ values are 0, 0.5, and 1, respectively, with the specific formula as follows:

$$\gamma = \begin{cases} 0, & C \geq 6, \\ 0.5, & 3 \leq C < 6, \\ 1, & C < 3, \end{cases} \quad (G = 8), \ C = \sum_{i=1}^{G} c_i \quad (1)$$

Here, $c_i$ indicates whether the answer of the rollout sample $o_i$ is correct, taking values of either 0 or 1. $C$ correspondingly represents the number of correct answers within a rollout group.

**GA$^2$DR.** The second approach is more characteristic of the audio modality. As mentioned earlier, when the model's attention to the audio segment is more dispersed, it suggests that the audio is complex and that the model struggles to identify a key point to solve the problem. In this case, the attention entropy is relatively large, corresponding to higher difficulty. Specifically, we use $a_j^{(n)}$, the attention after softmax from the last token position in the final hidden layer over all previous positions, and $\bar{p}_j$, the attention values assigned to audio tokens averaged across attention heads $N$, and then compute the entropy. This entropy is normalized across the batch to a value in [0,1], which directly represents difficulty. The complete calculation process is as follows:

$$a_j^{(n)} = \mathbf{A}_{T,j}^{(n)} \ , \ \bar{p}_j = \frac{1}{N} \sum_{n=1}^{N} a_j^{(n)} \ , \ H = -\sum \bar{p}_j \log \bar{p}_j \ , (j \in \mathcal{M}). \quad (2)$$

$$\gamma^{(b)} = \frac{H^{(b)} - \min_{b' \in \mathcal{B}} H^{(b')}}{\max_{b' \in \mathcal{B}} H^{(b')} - \min_{b' \in \mathcal{B}} H^{(b')}} \in [0, 1]. \tag{3}$$

Here, $\mathbf{A}$ denotes the complete attention matrix, $T$ represents the number of tokens, $\mathcal{M}$ indicates the indices corresponding to the audio attention part, $H$ denotes the computed entropy, and $\mathcal{B}$ is the batch size. The final $\gamma^{(b)}$ represents the difficulty value of the $b$-th sample within a batch. Compared with the first method, this approach provides a more fine-grained division of difficulty, no longer limited to three categories, but instead yielding a continuous difficulty range.

## 3.2 DIFFICULTY-ADAPTIVE LENGTH-BASED REWARD

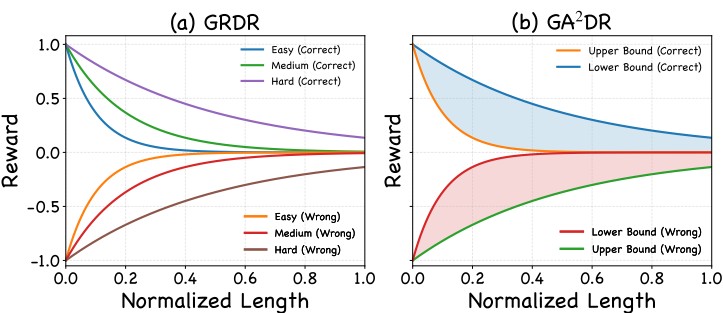

Figure 1: Curves of GRDR and GA$^2$DR with normalized length.

After defining question difficulty, the next step is to link difficulty with reasoning length in an effective way. Our core idea is short reasoning for simple questions and long reasoning for difficult ones, with reward values that change dynamically rather than staying fixed. Thus, we design a rule-based reward using a negative exponential function, with the corresponding curve shown in Figure 1. Specifically, it smoothly adjusts reward values according to the ratio between the current reasoning length and the model's maximum reasoning length. The decision between reward and penalty also depends on correctness. Incorrect samples always receive a penalty that decreases as reasoning length grows, encouraging further reasoning. Once a sample is answered correctly at a certain stage, the penalty switches to a positive reward, with the reward value increasing as reasoning length shortens, encouraging concise reasoning that retains only the core solution steps. For difficulty, specifically in GRDR (plot a), the difficulty levels are divided into three categories corresponding to three curves. Along the correct reward axis, difficulty increases from bottom to top, while on the wrong axis, it decreases accordingly. In contrast, GA$^2$DR (plot b) does not fix discrete difficulty levels but allows the curve exponent to vary continuously within the region. Both follow the same trend in which simple questions correspond to steeper curves and difficult ones to flatter curves, meaning that under the same reward value, difficult questions lead to longer reasoning lengths, and under the same reasoning length, simple questions yield smaller rewards. This achieves short reasoning for simple tasks and longer reasoning for difficult ones. The detailed calculation is as follows:

$$r_i = sign(o_i) \cdot e^{-k(\gamma)l_{o_i}} \tag{4}$$

$$k(\gamma) = (1 - \gamma)k_{easy} + \gamma k_{hard} \tag{5}$$

Here, the $sign$ function indicates whether a sample is correct or incorrect, taking only values 1 or -1. $l_{o_i}$ is the ratio of the sample's output length to the model's maximum output length, ranging from [0,1]. $k(\gamma)$ is obtained by applying linear interpolation to map the difficulty standards defined in the first part onto optimization curves with different slopes.

## 4 EXPERIMENTS

**Data.** For training, we use FS in this section, which is approximately twice the size of AVQA. The detailed data distribution has been presented in Section 2.

**Setup.** In this section, our experiments are mainly based on Qwen2-Audio-7B-Instruct and Qwen2.5-Omni-7B. All models are fine-tuned using LoRA with Prompt2. Most evaluations are conducted on MMAU-test-mini with ACC as the metric. And we also evaluate our proposed methods on MMAU-test-mini (v05.15.25), which improves the Q&A formulation and enhances the quality of the audio itself compared to the previous version, on MMAR (Ma et al., 2025), a benchmark designed to assess the deep reasoning capabilities of Audio Language Models (ALMs) in complex settings that span large-scale multitask, multimodal, and multilingual scenarios. Other details can be found in Appendix A.2.

Table 3: The performance of models on MMAU-test-mini with different base models and reward settings, reported under both the human-perspective difficulty annotations and the assigned model-perspective difficulty annotations. Here, † denotes the model-perspective difficulty annotations and ∗ denotes a leading proprietary model.

| Models | MMAU-Test-Mini | | | | | | | | | |
| | Sound | Music | Speech | Easy | Easy† | Medium | Medium† | Hard | Hard† | Avg |
| --- | --- | --- | --- | --- | --- | --- | --- | --- | --- | --- |
| Baseline Models | | | | | | | | | | |
| Qwen2-Audio-7B-Instruct | 53.75 | 48.80 | 47.74 | 48.06 | 62.80 | 50.58 | 45.32 | 51.29 | 28.18 | 50.10 |
| Qwen2.5-Omni-7B | 67.26 | 59.88 | 53.75 | 54.26 | 90.89 | 71.96 | 44.39 | 41.37 | 11.19 | 60.30 |
| Kimi-Audio-7B-Instruct | 72.37 | 58.98 | 61.66 | 50.38 | 91.84 | 75.88 | 52.80 | 54.74 | 18.14 | 64.40 |
| Gemini2.5-Pro-0506[∗1] | 70.57 | 65.26 | 62.16 | 52.32 | 95.82 | 77.64 | 57.47 | 55.60 | 12.35 | 66.00 |
| Based On Qwen2-Audio-7B-Instruct | | | | | | | | | | |
| GRPO | 69.37 | 60.48 | 55.26 | 49.61 | 82.82 | 71.76 | 47.66 | 53.02 | 30.16 | 61.70 |
| + TR | 68.16 | 60.77 | 55.85 | 48.83 | 83.87 | 71.56 | 46.26 | 53.87 | 28.95 | 61.60 |
| + GRDR | 66.96 | 58.38 | 60.06 | 54.26 | 81.59 | 69.60 | 45.32 | 53.01 | 35.13 | 61.80 |
| Based On Qwen2.5-Omni-7B | | | | | | | | | | |
| GRPO | **73.57** | 65.56 | **69.06** | 59.69 | **93.92** | 78.23 | 59.81 | **60.77** | 27.41 | 69.40 |
| + TR | 72.97 | 66.46 | 65.16 | 58.14 | 93.73 | 78.43 | 57.47 | 57.75 | 25.86 | 68.40 |
| + GRDR | 71.47 | **72.45** | 66.66 | **60.07** | 93.16 | 80.00 | 58.87 | 59.91 | **32.81** | **70.20** |
| + GA$^2$DR | 71.47 | 71.85 | 66.66 | 57.75 | 92.78 | **80.58** | **59.81** | 60.34 | 32.04 | 70.00 |

Table 4: Evaluations on MMAU-v0515. Here, ∗ denotes a leading proprietary model.

Table 5: Evaluations on MMAR. Here, ∗ denotes a leading proprietary model.

| Models | MMAU-Test-Mini-v0515 | | | |
| | Sound | Music | Speech | Avg |
| --- | --- | --- | --- | --- |
| Baseline Models | | | | |
| Qwen2-Audio | 62.16 | 62.27 | 55.55 | 60.00 |
| Qwen2.5-Omni | 74.17 | 65.26 | 61.56 | 67.00 |
| Kimi-Audio | 78.97 | 60.47 | 66.96 | 68.80 |
| Gemini2.5-Pro∗ | 76.57 | 73.95 | 80.78 | 77.10 |
| Based On Qwen2.5-Omni-7B | | | | |
| GRPO | 84.08 | 69.46 | 74.17 | 75.90 |
| + TR | 83.78 | 70.65 | 74.47 | 76.30 |
| + GRDR | 83.48 | 70.35 | 75.97 | 76.60 |
| + GA$^2$DR | 83.18 | 71.55 | 75.67 | **76.80** |

| Models | MMAR | | | |
| | Sound | Music | Speech | Avg |
| --- | --- | --- | --- | --- |
| Baseline Models | | | | |
| Qwen2-Audio | 33.33 | 24.27 | 32.31 | 30.00 |
| Qwen2.5-Omni | 58.79 | 40.78 | 59.86 | 56.70 |
| Kimi-Audio | 57.57 | 45.63 | 63.26 | 59.00 |
| Gemini2.5-Pro∗ | 73.33 | 64.07 | 88.77 | 80.50 |
| Based On Qwen2.5-Omni-7B | | | | |
| GRPO | 60.00 | 48.05 | 62.24 | 59.90 |
| + TR | 64.84 | 49.51 | 63.94 | 61.90 |
| + GRDR | 61.21 | 51.94 | 65.30 | 61.20 |
| + GA$^2$DR | 64.84 | 54.85 | 65.30 | **62.90** |

## 4.1 MAIN RESULTS

Table 3 mainly presents three parts: the performance of four baseline models and our proposed rewards on Qwen2-Audio-7B-Instruct and Qwen2.5-Omni-7B. We further report results on MMAU-

---

[1]https://ai.google.dev/gemini-api/docs/models#gemini-2.5-pro

v0515 and MMAR, together with the base model on these benchmarks, shown in Table 4 and Table 5. TR denotes the basic Truncation Reward (Liu et al., 2025), with its formula provided in Appendix A.2. In addition, we extend MMAU-test-mini with model-perspective difficulty labels annotated by the four baseline models in Table 3, with details given in Appendix A.3.

**Performance across Models.** First, we analyze the four baseline models. Their overall performance reflects different capability levels and the typical range of most systems. However, they differ markedly on hard questions. Even Gemini2.5-Pro, which is nearly the strongest on average, performs poorly on hard items, while Qwen2-Audio, the weakest on average, shows a clear advantage. We attribute this to the combined effects of the LLM backbone strength and the number of supported modalities. For easy questions, the LLM backbone is dominant because these items require minimal audio understanding and rely mainly on text comprehension. In contrast, hard questions demand stronger audio interpretation, where pure LALMs often outperform Omni-style models that integrate more modalities. This pattern is evident in Qwen2.5-Omni and Kimi-Audio, which share a "Whisper + Qwen2.5" architecture but still exhibit a noticeable gap on hard questions.

Second, we compare our proposed GRDR and GA$^2$DR with TR. For Qwen2-Audio-7B-Instruct, the three methods achieve similar overall performance, with GRDR performing best and TR worst, confirming the effectiveness of our approach. TR shows slightly better performance on medium questions, but GRDR is clearly stronger on hard ones. For Qwen2.5-Omni-7B, both of our methods deliver clear overall gains. They substantially outperform TR on medium and hard questions, while maintaining comparable results on easy ones. This shows that our methods effectively utilize question difficulty to assign appropriate rewards, promote deeper reasoning on hard questions, and improve performance on challenging tasks. Unlike TR, which uses a fixed truncation length and only constrains part of the samples, our reward designs offer more balanced treatment across difficulty levels, leading to stronger overall results.

Third, we compare GRDR and GA$^2$DR, focusing on the results based on Qwen2.5-Omni-7B. The two methods differ in that GRDR is outcome-oriented, whereas GA$^2$DR is process-oriented. GRDR performs better on easy questions, GA$^2$DR is stronger on medium ones, and their performance is similar on hard questions. We believe this is due to their different difficulty definitions: GRDR uses only three levels, while GA$^2$DR applies a finer, unconstrained difficulty scale, which particularly benefits medium questions. For easy and hard questions, GA$^2$DR is slightly weaker because normalization may place samples with similar audio attention entropy into different difficulty bins, diminishing its advantage.

In summary, both GRDR and GA$^2$DR achieve clear performance gains, especially on hard questions, showing that our methods can effectively adapt to different questions according to their difficulty.

**Performance across Benchmarks.** To further evaluate our model in broader and more reasoning-intensive scenarios, we conduct additional evaluations on MMAU-v0515 and MMAR, which also help verify the generalization ability of our methods. In terms of overall performance, our two methods still achieve the best results on MMAU-v0515 compared with all baselines. On MMAR, however, the two methods behave differently: GA$^2$DR maintains a clear performance lead, whereas GRDR falls behind the TR method.

This difference can be explained by the nature of the two approaches. GRDR is outcome-oriented, while GA$^2$DR is process-oriented. GRDR is highly susceptible to noise in rollout samples, making it prone to reward hacking. As a result, it performs reasonably well on the relatively simpler MMAU benchmarks but degrades significantly on the more challenging MMAR benchmark. In contrast, GA$^2$DR is unaffected by such noise because it determines difficulty solely based on the model's current behavior and the audio characteristics of each question. This removes much of the randomness and reduces the likelihood of reward hacking, leading to more stable performance across benchmarks and consistently strong results on more challenging tasks. We also conduct additional experiments to further verify generalization. The results confirm that GRDR is indeed affected by random noise introduced during rollout; however, once this issue is mitigated through appropriate constraints, its performance still surpasses all baseline methods. Detailed results and analysis are provided in Appendix A.4.

## 4.2 Analysis of Reasoning Length across Models

In this subsection, we mainly compare reasoning length from both difficulty perspectives, as shown in Figure 2, focusing on the four baseline models and all models based on Qwen2.5-Omni-7B. Because the length gaps between models are relatively large, we present log-scaled lengths in this part. We also provide Figure 3, which shows three Qwen2.5-Omni-7B–based models—direct GRPO and our two proposed rewards. To better illustrate how our methods adjust reasoning length across different difficulty levels, this figure uses the actual token counts without log-scaling. In addition, detailed length statistics are provided in Appendix A.5.

First, examining length trends under the two perspectives of difficulty, clear differences emerge. Under the human perspective, reasoning is longer for easy and hard questions, with medium questions shortest, whereas under the model perspective, reasoning length increases with difficulty. This indicates a fundamental inconsistency between human and model standards. Consequently, using human-perspective difficulty in training may conflict with the model's perspective. Furthermore, since different models within the same perspective already show varying trends, models at different training stages naturally behave like distinct models, each with its own patterns. Together, these observations strongly support adopting model-perspective difficulty in training and continuously updating the difficulty standard as the model improves.

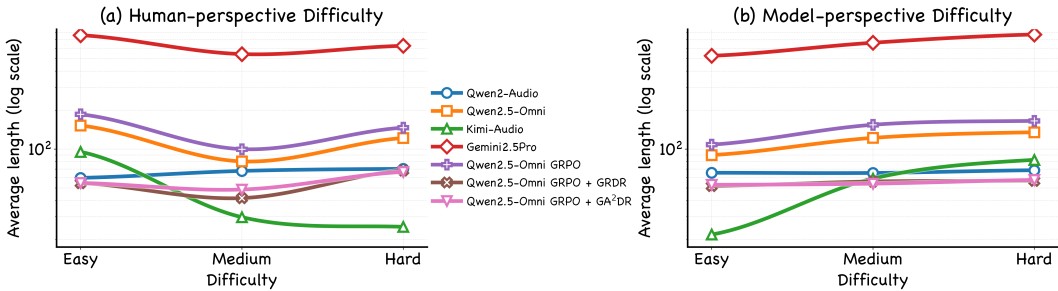

Figure 2: The trend of average length across different models on MMAU-Test-Mini, under both the human-perspective difficulty and model-perspective difficulty. The length is measured in tokens and is presented after applying a logarithmic transformation.

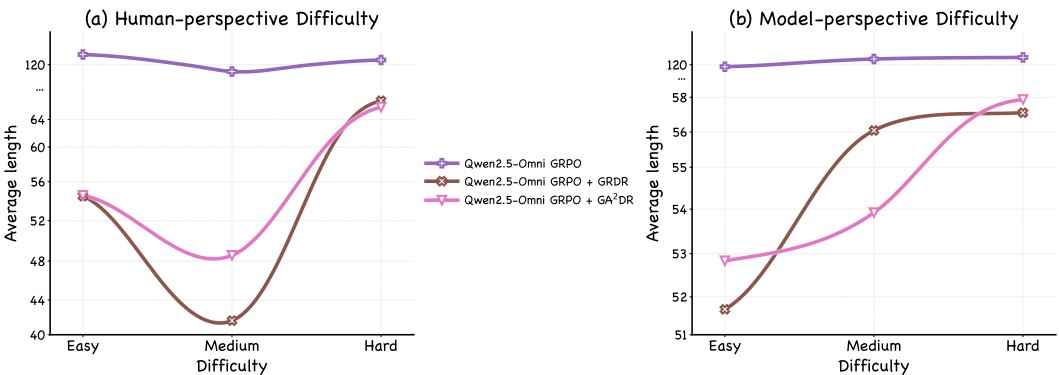

Figure 3: The trend of average reasoning length for direct GRPO and our two proposed methods on MMAU-Test-Mini, evaluated under both human-perspective and model-perspective difficulty. Length is measured directly in tokens without any logarithmic transformation.

Second, when comparing reasoning lengths of our proposed rewards with other models, we find that our two rewards produce shorter reasoning across all difficulty levels than most other models. Under the human perspective, the lengths of our methods are close to direct GRPO for easy and hard questions but are much shorter on medium questions. Under the model perspective, the curves of our two rewards almost overlap and show clear improvements over both the base model and direct GRPO at all difficulty levels. Moreover, under the model perspective, reasoning length increases

appropriately with difficulty. These results strongly demonstrate the effectiveness of our proposed rewards, achieving shorter reasoning for easy questions and deeper reasoning for hard ones.

In summary, our proposed difficulty-adaptive length-based reward is both reasonable and effective. Specifically, it achieves short reasoning for simple questions and long reasoning for difficult ones, while overall significantly reducing reasoning length. This makes LALMs reason smartly, achieving greatly improved reasoning efficiency alongside higher performance.

### 4.3 Ablation study: Results of single length-based reward and $k$ settings

In this section, we conduct an in-depth ablation study on our proposed difficulty-adaptive length-based reward. Specifically, we remove the mechanism in Equation 5 that determines different reward curves based on question difficulty, and replace $k(\lambda)$ in Equation 4 with a fixed value $k$, meaning that all questions share the same reward curve. We also experiment with different settings of $k$ to further validate the effectiveness of our approach. The main evaluation is performed on MMAU-Test-Mini in Table 6, and additional results on MMAU-v0515 and MMAR are also provided in Table 7 and Table 8.

Table 6: The performance of models trained with the single length-based reward setting. Here, † denotes the model-perspective difficulty annotations.

| Models | MMAU-Test-Mini | | | | | | | | | |
| | Sound | Music | Speech | Easy | Easy$^\dagger$ | Medium | Medium$^\dagger$ | Hard | Hard$^\dagger$ | Avg |
| GRPO Based On Qwen2.5-Omni-7B (With Length-based Reward Only) | | | | | | | | | | |
| k=2 | 73.27 | 67.06 | 67.86 | 59.69 | 94.49 | 77.84 | 58.41 | 61.63 | 27.41 | 69.40 |
| k=6 | 72.37 | 66.46 | 68.76 | 56.20 | 93.92 | 79.21 | 57.00 | 61.63 | 28.95 | 69.20 |
| k=10 | 72.37 | 70.06 | 66.36 | 58.91 | 93.92 | 79.60 | 56.07 | 59.48 | 31.27 | 69.60 |

Table 7: Evaluations on MMAU-v0515.

| Models | MMAU-Test-Mini-v0515 | | | |
| | Sound | Music | Speech | Avg |
| GRPO Based On Qwen2.5-Omni-7B | | | | |
| k=2 | 81.38 | 73.05 | 74.77 | 76.40 |
| k=6 | 82.88 | 71.55 | 74.77 | 76.40 |
| k=10 | 81.98 | 71.25 | 76.57 | 76.60 |

Table 8: Evaluations on MMAR.

| Models | MMAR | | | |
| | Sound | Music | Speech | Avg |
| GRPO Based On Qwen2.5-Omni-7B | | | | |
| k=2 | 61.81 | 50.97 | 63.26 | 61.40 |
| k=6 | 60.60 | 52.91 | 64.96 | 61.80 |
| k=10 | 61.81 | 52.91 | 64.62 | 62.20 |

From the perspective of overall performance, all fixed k-value settings outperform direct GRPO and the traditional TR method. This demonstrates the effectiveness of our length-based reward and shows that these samples indeed require length optimization—direct GRPO tends to produce either redundant or insufficient reasoning. It also indicates that, compared with TR, our reward design can dynamically adjust rewards based on length rather than relying on a fixed threshold, resulting in better optimization. Comparing with the difficulty-adaptive methods GRDR and GA$^2$DR, most fixed k settings perform worse, further confirming the validity and effectiveness of incorporating adaptive difficulty. Across the two MMAU benchmarks, performance fluctuates noticeably across all three difficulty levels under different k values, showing that applying a single reward curve to all questions leads to imbalance and reduced overall performance. Overall, these experiments further validate the effectiveness of our difficulty-adaptive length-based reward.

## 5 Related work

In recent years, advances in LLMs have driven the development of MLLMs, enhancing multimodal understanding. In the audio domain, LALMs such as Qwen2-Audio (Chu et al., 2024), Audio Flamingo (Kong et al., 2024), and SALMONN (Tang et al., 2024) handle basic understanding well

but remain limited on complex tasks due to short outputs and lack of reasoning. Later models like Qwen2.5-Omni (Xu et al., 2025) and Kimi-Audio (KimiTeam et al., 2025) demonstrate some initial reasoning ability but still rely heavily on SFT, leaving outputs fixed and dependent on pretraining data, thus performing poorly in complex scenarios. To address this, some studies extend SFT, such as Audio-Reasoner (Xie et al., 2025), which uses large-scale CoT-annotated pairs to achieve reasoning via SFT. Others adopt RL, as in R1-AQA (Li et al., 2025) and Omni-R1 (Rouditchenko et al., 2025), which avoids using data with CoT, and promotes self-driven reasoning.

However, prior work mainly focuses on overall performance, leaving open key questions: how GRPO differs from SFT, and whether improvements come from explicit reasoning or from prompts activating implicit reasoning. Therefore, in this study, we conduct a systematic and deep analysis of these two questions to provide a clearer understanding of different approaches in LALMs. Based on this in-depth analysis, we draw an important conclusion: The explicit reasoning introduced by GRPO is necessary, especially for more difficult questions, but it often leads to redundant reasoning on simple ones. Therefore, reasoning length should be optimized according to problem difficulty, reducing redundancy for simple questions while encouraging deeper reasoning for harder ones.

Reasoning efficiency thus becomes a major challenge. Prior studies (Qu et al., 2025) highlight inefficiencies such as redundant content and overthinking. RL-based methods explore reward function designs to address this issue (Arora & Zanette, 2025; Aggarwal & Welleck, 2025; Shen et al., 2025). Most methods rely on setting a fixed length threshold for optimization. On the one hand, such approaches do not link reasoning length with problem type and therefore cannot adapt to all kinds of problems. On the other hand, the fixed threshold cannot change with the model's evolving ability during training. Liu et al. (2025) summarizes these efforts and proposes a difficulty-aware dynamic approach that improves both performance and efficiency. However, this method depends on an additional independent dataset during training to measure the model's capability, which not only increases computational cost but also makes the results sensitive to how this dataset is selected. In LALMs, Audio-Thinker (Wu et al., 2025) instead tackles the problem through a "when to think" mechanism, dividing tasks into those requiring reasoning and those that do not, but it does not further optimize the samples that require reasoning. In this work, we propose two difficulty-adaptive length-based rewards to enable efficient and effective reasoning in LALMs. Our method maintains comparable or even superior performance while reducing overall reasoning length, encouraging concise reasoning for simple questions, deeper reasoning for difficult ones, and progressive optimization as the model's capability improves.

## 6 LIMITATIONS AND FUTURE WORK

In this work, we optimize reasoning across difficulty levels and achieve notable gains. But, limitations still remain. Our analysis of outputs from all models—including prior work and proprietary models—indicates that a strong CoT response should extract key information, perform structured reasoning, and deliver a clear final answer. Detailed analyses and examples are provided in Appendix A.9 and Appendix A.10. We also believe that incorporating rewards from external LLM APIs may further enhance the coherence and reliability of CoT outputs while preserving the intended reasoning paradigm, which we will explore in future work.

## 7 CONCLUSION

In this work, we focus on addressing the question of how LALMs can achieve reasoning more efficiently and effectively. First, we conduct detailed experimental analyses on two key issues: in which conditions SFT and GRPO are more effective, and whether performance improvements come from explicit prompts directly or from implicit prompts activating the model's reasoning ability. Our findings show that explicit reasoning through GRPO is more effective, but the reasoning length should be optimized according to question difficulty. Based on this, we propose two difficulty-adaptive length-based rewards and carry out extensive experiments. The results demonstrate that our method achieves better overall performance, significantly shortens reasoning length, and improves efficiency. We also provide qualitative analyses of reasoning outputs from multiple models, identify an ideal reasoning structure paradigm, and recommend a set of training procedures for models with different capability levels, offering useful guidance for future work.

## ETHICS STATEMENT

This work adheres to the ICLR Code of Ethics. In this study, no human subjects or animal experimentation is involved. All datasets used are sourced in compliance with relevant usage guidelines, ensuring no violation of privacy. We take care to avoid any biases or discriminatory outcomes in our research process. No personally identifiable information is used, and no experiments are conducted that raise privacy or security concerns. We remain committed to maintaining transparency and integrity throughout the research process.

## REPRODUCIBILITY STATEMENT

In this section, we provide a detailed description of the reproduction details of this work.

Our base models, training framework, and datasets are all publicly available, with detailed information provided in Appendix A.2. In addition, we provide an anonymous GitHub repository https://anonymous.4open.science/r/ICLR2026-Anonymous-Repo-24196/ to support the reproduction of this work. The repository contains scripts for the two proposed difficulty-adaptive length-based rewards in this paper, as well as the baseline TR method, which can be directly integrated into the ms-swift framework. It also includes all evaluation scripts and reasoning-length assessment scripts, the Cold-Start dataset used in our experiments (including both audio wav files and JSONL annotations), and the outputs of different models on the 20 sarcasm cause detection questions discussed in the analysis.

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

## A APPENDIX

### A.1 TASK DEFINITION

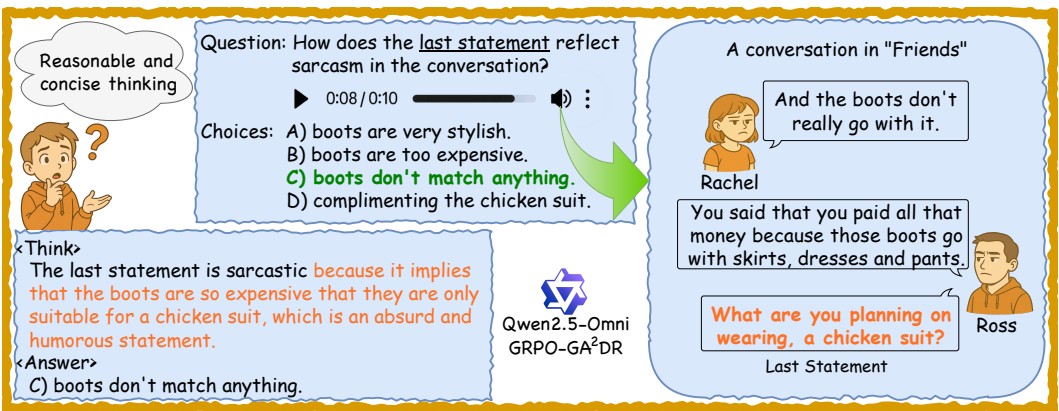

Figure 4: An audio QA example from "*Friends*". The top-left shows the question and options (green indicates the correct one), the right side presents the audio dialogue, and the bottom-left shows the output of our proposed method on Qwen2.5-Omni-7B.

### A.2 SETUP

**Models.** For the base models, we primarily adopt Qwen2-Audio-7B-Instruct and Qwen2.5-Omni-7B, which are among the most widely used open-source LALMs. These two models are also frequently used in prior work, facilitating fair and direct comparisons. For the training framework, we leverage the ms-swift (Zhao et al., 2025) and perform training on three A100-40G GPUs, where two GPUs are used for model training and one GPU is reserved for vLLM-based inference (Kwon et al., 2023). More training details and hyper-parameters can be found in the Appendix A.8.

**Datasets.** For the training datasets, we use two in total: FS and AVQA (Yang et al., 2022). The latter is a subset of the former, while FS is constructed by augmenting AVQA with four additional datasets. For AVQA, we only keep the audio–text pairs and replace the word "video" in the questions with "audio." For evaluation, we mainly test on MMAU-test-mini (Sakshi et al., 2025), with AirBench Foundation-Sound (Yang et al., 2024) as a secondary reference.

**GRPO.** GRPO has been extensively applied in both LLMs and MLLMs, achieving notable progress, and our implementation largely follows prior studies (DeepSeek-AI et al., 2025; Li et al., 2025). Compared with other RL methods, the key feature of GRPO is that it evaluates the policy model's advantage using the average reward of in-group sampled outputs. Given an input question, a set of sampled responses for that question, and their rewards from the reward function, the advantage is calculated as follows:

$$A_i = \frac{r_i - mean(\{r_1, r_2, \cdots, r_G\})}{std(\{r_1, r_2, \cdots, r_G\})}. \tag{6}$$

Here, $A_i$ denotes the advantage used to optimize the policy model, $\{r_1, r_2, \cdots, r_G\}$ represents the set of reward values corresponding to each sampled output within the group $\{o_1, o_2, \cdots, o_G\}$, and $G$ indicates the number of samples in the group.

After this, GRPO uses the computed advantage to optimize the policy model by maximizing the following objective function:

$$\mathcal{J}_{\text{GRPO}}(\theta) = \mathbb{E}[q \sim P(Q), \{o_i\}_{i=1}^{G} \sim \pi_{\theta_{old}}(O|q)]$$

$$\frac{1}{G} \sum_{i=1}^{G} \left( \min \left( \frac{\pi_\theta(o_i|q)}{\pi_{\theta_{old}}(o_i|q)} A_i, \text{clip} \left( \frac{\pi_\theta(o_i|q)}{\pi_{\theta_{old}}(o_i|q)}, 1 - \varepsilon, 1 + \varepsilon \right) A_i \right) - \beta \mathcal{D}_{KL}(\pi_\theta || \pi_{ref}) \right). \tag{7}$$

$$\mathbb{D}_{\text{KL}}(\pi_\theta \,\|\, \pi_{\text{ref}}) = \frac{\pi_{\text{ref}}(o_i \mid q)}{\pi_\theta(o_i \mid q)} - \log \frac{\pi_{\text{ref}}(o_i \mid q)}{\pi_\theta(o_i \mid q)} - 1. \tag{8}$$

Here, $\epsilon$ and $\beta$ are hyper-parameters.

**Cold-start GRPO.** A common way to learn the reasoning structure of advanced models and further optimize performance is to distill their outputs and then apply SFT to quickly teach the model such paradigms. After this, GRPO is performed, and this approach is referred to as Cold-Start. To explore the effectiveness of this method, we also conduct experiments with SFT Cold-Start followed by GRPO. For the Cold-Start dataset, we first sample a subset from the FS training set, distill it with Gemini2.5Pro, and then use Qwen3-235B-A22B[3] to retain only samples that are both correct and consistent in reasoning and answers. From these, we select 200 per task to form a dataset of 1,000 samples. This dataset is then used to perform Cold-Start SFT on Qwen2-Audio-7B-Instruct, and the model obtained after 2 epochs of SFT serves as the starting checkpoint for GRPO.

**Implicit and Explicit Prompt.** To examine whether the effectiveness of CoT comes from explicit outputs or simply triggering implicit reasoning, we design two prompts, shown in Figure 5. For each sample, the model generates its answer within `<answer> </answer>`. In Prompt1, CoT is not required, while in Prompt2, it is generated within `<think> </think>`. A Format-Reward enforces this structure in Prompt2, whereas in Prompt1 it only regulates the answer format.

> **Prompt Templates**
>
> Prompt1:Output the most suitable answer (from A, B, C, and D with its corresponding answer) in <answer> </answer> tags.
>
> Prompt2:Output your thinking process in <think> </think> and put the most suitable answer (from A, B, C, and D with its corresponding answer) in <answer> </answer> tags.

Figure 5: Different prompt templates for GRPO, where Prompt1 is the implicit prompt and Prompt2 is the explicit prompt.

---

[3]https://huggingface.co/Qwen/Qwen3-235B-A22B

**Truncation Reward.**     In addition to our designed rewards, we also consider the most straightforward way to control reasoning length as one of the baseline methods. This strategy sets a length threshold $L_T$, where rollout samples that are correct and have length less than or equal to $L_T$ receive a positive reward, while those exceeding it receive a penalty, even if the answer is correct. The calculation is as follows:

$$r_i = \begin{cases} 1, & \text{if } L_{o_i} \leq L_T \text{ and } o_i \text{ is correct,} \\ \sigma, & \text{if } L_{o_i} > L_T. \end{cases} \tag{9}$$

Here, $L_{o_i}$ denotes the output length of the CoT for the corresponding sample, and $\sigma$ is a penalty hyper-parameter.

### A.3    MODEL-PERSPECTIVE DIFFICULTY ON MMAU

In Sakshi et al. (2025), question difficulty was annotated manually, where multiple experts assigned difficulty scores to each question. These labels are of very high quality, but the cost of such annotation is prohibitively high, and the labels are fixed, making it difficult to align them with the model's evolving state across different training steps.

Therefore, in this subsection, we introduce model-perspective difficulty labels. Since each model has different capabilities, we aim to reflect an average level across models. To this end, we adopt four different models: Qwen2-Audio-7B-Instruct, Qwen2.5-Omni-7B, Kimi-Audio-7B-Instruct[4], and Gemini2.5-Pro-0506. These models perform inference on MMAU-test-mini under the same random seed, and difficulty labels from the model's perspective are assigned based on the number of models answering each question correctly. The detailed distribution of model-perspective difficulty labels is shown in Table 9.

Table 9: Data distribution of difficulty from human (Orig) and model perspectives, including counts of changed (Chg) and unchanged (Un-Chg) samples, and transitions across difficulty categories.

| Orig Diff. | Total-Num Orig | Total-Num New | Num Un-Chg | Num Chg | New Diff. | Chg Num |
|---|---|---|---|---|---|---|
| Easy | 258 | 527 | 97 | 161 | Medium | 68 |
| | | | | | Hard | 93 |
| Medium | 510 | 214 | 91 | 419 | Easy | 338 |
| | | | | | Hard | 81 |
| Hard | 232 | 259 | 85 | 147 | Easy | 92 |
| | | | | | Medium | 55 |

### A.4    IMPACT OF THRESHOLD RATIO ON OUR PROPOSED REWARD

Considering that when the negative exponential function approaches 1 its exponent tends toward 0, this can represent a form of implicit reasoning, though it may reduce readability. To further analyze this issue, we introduce a parameter $l_{min}$, which serves as the minimum threshold ratio. When the relative length of a sample is smaller than this value, it is directly set to 1; when it is larger, its length is further normalized. The $\zeta$ then applies this secondary normalization after the threshold.

$$r_i = sign(o_i) \cdot e^{-k(\gamma)\zeta(l_{o_i}; l_{min})} \tag{10}$$

$$\zeta(l_{o_i}; l_{min}) = max(0, \frac{l_{o_i} - l_{min}}{1 - l_{min}}) \tag{11}$$

Table 10, Table 11, and Table 12 report the effects of adding a threshold ratio to our two proposed rewards on three benchmarks. Models without special notation correspond to the case of $l_{min} = 0$, which means that no threshold ratio is applied. For the cases with $l_{min} = 0.1$, this value is determined based on the average reasoning length of direct GRPO models.

---

[4]https://huggingface.co/moonshotai/Kimi-Audio-7B-Instruct

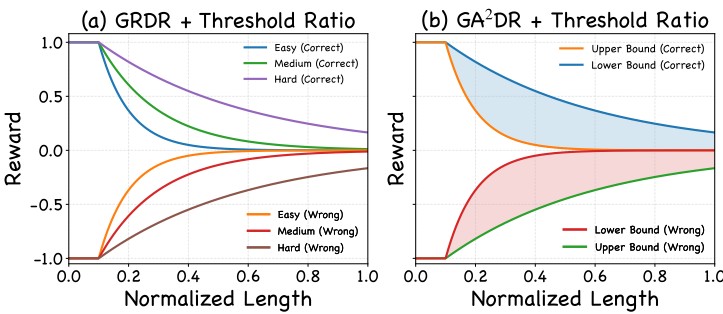

Figure 6: Curves of GRDR and GA$^2$DR with normalized length and the threshold ratio $l_{min}$.

Table 10: The performance of our proposed GRDR, GA$^2$DR, and their variants with added length threshold ratios on MMAU-Test-Mini. Here, unmarked models correspond to $l_{min} = 0$, and † denotes the model-perspective difficulty annotations.

| Models | MMAU-Test-Mini | | | | | | | | | |
|---|---|---|---|---|---|---|---|---|---|---|
| | Sound | Music | Speech | Easy | Easy† | Medium | Medium† | Hard | Hard† | Avg |
| Based On Qwen2.5-Omni-7B | | | | | | | | | | |
| GRDR | 71.47 | **72.45** | 66.66 | **60.07** | 93.16 | 80.00 | 58.87 | 59.91 | **32.81** | **70.20** |
| + $l_{min} = 0.1$ | **72.07** | 69.76 | 63.36 | 55.81 | **93.92** | 79.60 | 55.14 | 57.75 | 27.41 | 68.40 |
| GA$^2$DR | 71.47 | 71.85 | **66.66** | 57.75 | 92.78 | **80.58** | **59.81** | 60.34 | 32.04 | 70.00 |
| + $l_{min} = 0.1$ | 71.77 | 68.86 | 64.26 | 55.42 | 93.73 | 79.02 | 57.00 | 59.05 | 25.86 | 68.20 |

Table 11: The performance of our proposed rewards, and their variants with added length threshold ratios on MMAU-v0515. Here, unmarked models correspond to $l_{min} = 0$.

Table 12: The performance of our proposed rewards, and their variants with added length threshold ratios on MMAR. Here, unmarked models correspond to $l_{min} = 0$.

| Models | MMAU-Test-Mini-v0515 | | | |
|---|---|---|---|---|
| | Sound | Music | Speech | Avg |
| Based On Qwen2.5-Omni-7B | | | | |
| GRDR | **83.48** | 70.35 | 75.97 | 76.60 |
| + $l_{min} = 0.1$ | 83.18 | 70.95 | 76.27 | 76.80 |
| GA$^2$DR | 83.18 | 71.55 | 75.67 | 76.80 |
| + $l_{min} = 0.1$ | 81.08 | **72.15** | **77.77** | **77.00** |

| Models | MMAR | | | |
|---|---|---|---|---|
| | Sound | Music | Speech | Avg |
| Based On Qwen2.5-Omni-7B | | | | |
| GRDR | 61.21 | 51.94 | 65.30 | 61.20 |
| + $l_{min} = 0.1$ | 63.63 | 52.91 | **65.98** | 63.00 |
| GA$^2$DR | 64.84 | **54.85** | 65.30 | 62.90 |
| + $l_{min} = 0.1$ | **64.84** | 53.39 | 63.60 | **63.00** |

From an overall performance perspective, models with a length threshold ratio achieve higher average scores on MMAU-Test-Mini-v0515 and MMAR, but show the opposite trend on MMAU-Test-Mini. Considering that the quality of the Q&As and audio files in MMAU-Test-Mini is relatively poor, the performance on the latter two benchmarks is therefore more convincing. The fact that introducing a length threshold leads to better results on most benchmarks further indicates that our length-based reward curve tends to infinitely optimize toward a completion length of zero, which may cause the CoT content of certain samples to become less effective, ultimately resulting in a decrease in overall performance.

From the perspective of different difficulty levels, this part mainly focuses on the results from the two versions of MMAU-Test-Mini. We observe that the model with the added threshold performs better on easy and hard questions, while the model without the threshold achieves better results on medium questions. This outcome is closely related to our threshold setting: the threshold value was determined based on the average reasoning length of the direct GRPO model on MMAU-Test-Mini. Since more than half of the questions in the dataset are labeled as medium difficulty, the threshold essentially aligns with this group. As a result, medium questions— which could have

benefited from further length optimization—received less optimization, leading to relatively lower performance. In contrast, for easy and hard questions, introducing the threshold brings two benefits: for easy questions, it prevents excessively short reasoning that may reduce CoT effectiveness; for hard questions, it increases the upper bound of reasoning length, enabling deeper exploration of problem-solving strategies. Consequently, the model achieves better performance on both easy and hard questions.

In addition, when focusing on GRDR and GA$^2$DR, we can observe that the latter achieves better performance on two benchmarks. This is because GA$^2$DR defines difficulty levels at the batch level, which aligns with the batch-wise optimization process during model backpropagation, making this approach more effective. Moreover, its attention mechanism jointly considers both the textual question and the characteristics of the audio itself.

Furthermore, previous studies (Peng et al., 2025) have shown that models trained with reinforcement learning tend to exhibit improvements in pass@1 but declines in pass@k, largely due to the instability of rollout samples. Our GA$^2$DR method effectively mitigates this issue, since all rollout samples within a group share the same question and audio input, thereby avoiding inconsistencies caused by rollout variance.

In summary, combining GA$^2$DR with a length threshold provides the most balanced and effective optimization strategy. The specific threshold value, however, should be determined based on the characteristics of each individual task.

## A.5 DETAILED LENGTH STATISTICS OF DIFFERENT MODELS ON MMAU

In this section, we provide the raw reasoning-length data for all models used in our main experiments on MMAU-Test-Mini, without any post-processing. All lengths are measured using the Qwen2-Audio tokenizer. Note that Qwen2-Audio and Qwen2.5-Omni share the same tokenizer, so using the former does not introduce any error when evaluating the latter. For Gemini2.5, however, we cannot obtain its tokenizer because it is a closed-source model. Although using the Qwen2-Audio tokenizer may introduce some deviation, the reasoning lengths of Gemini2.5 are far greater than those of all our methods, so this small discrepancy does not affect the clearly observable relative differences.

Table 13: The output length statistics of models on MMAU-test-mini with different base models and reward settings, reported under both the human-perspective difficulty annotations and the assigned model-perspective difficulty annotations. Here, † denotes the model-perspective difficulty annotations.

| Models | MMAU-Test-Mini | | | | | |
|---|---|---|---|---|---|---|
| | Easy | Easy$^\dagger$ | Medium | Medium$^\dagger$ | Hard | Hard$^\dagger$ |
| Baseline Models | | | | | | |
| Qwen2-Audio-7B-Instruct | 59.79 | 65.48 | 67.66 | 65.26 | 70.18 | 68.50 |
| Qwen2.5-Omni-7B | 151.80 | 89.76 | 80.19 | 121.90 | 121.57 | 134.67 |
| Kimi-Audio-7B-Instruct | 94.99 | 21.79 | 29.69 | 59.03 | 25.01 | 82.38 |
| Gemini2.5-Pro-0506 | 759.60 | 525.85 | 542.17 | 663.58 | 628.32 | 768.81 |
| Based On Qwen2-Audio-7B-Instruct | | | | | | |
| GRPO | 41.76 | 42.26 | 42.55 | 43.29 | 43.89 | 42.97 |
| + TR | 42.11 | 43.55 | 43.77 | 44.42 | 46.32 | 44.31 |
| + GRDR | 27.76 | 30.41 | 31.59 | 31.49 | 32.26 | 30.88 |
| Based On Qwen2.5-Omni-7B | | | | | | |
| GRPO | 185.38 | 108.08 | 99.74 | 153.90 | 146.18 | 164.92 |
| + TR | 183.65 | 110.20 | 103.10 | 153.49 | 145.76 | 165.46 |
| + GRDR | 54.49 | 51.69 | 41.70 | 56.05 | 68.32 | 56.73 |
| + $l_{min} = 0.1$ | 124.62 | 89.10 | 85.66 | 110.00 | 107.54 | 116.96 |
| + GA$^2$DR | 54.58 | 52.84 | 48.57 | 53.92 | 66.48 | 57.58 |
| + $l_{min} = 0.1$ | 115.38 | 88.80 | 85.34 | 105.89 | 105.91 | 109.68 |

### A.6 REASONING LENGTH VS. ACCURACY

In this section, we analyze the relationship between output length and accuracy for the GRDR and GA$^2$DR methods on MMAU-Test-Mini. From Figure 7 to Figure 10, samples are grouped into length intervals of similar size, and correctly answered samples within each interval are further divided by difficulty level, indicated by increasingly darker shades of blue.

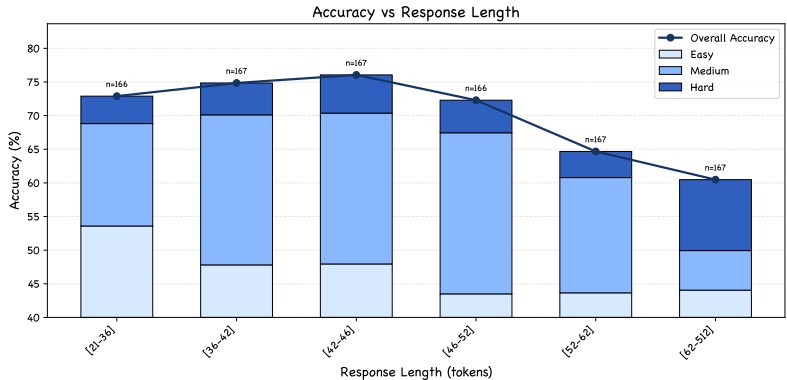

Figure 7: The trend of length and accuracy for the GRDR with human-perspective difficulty.

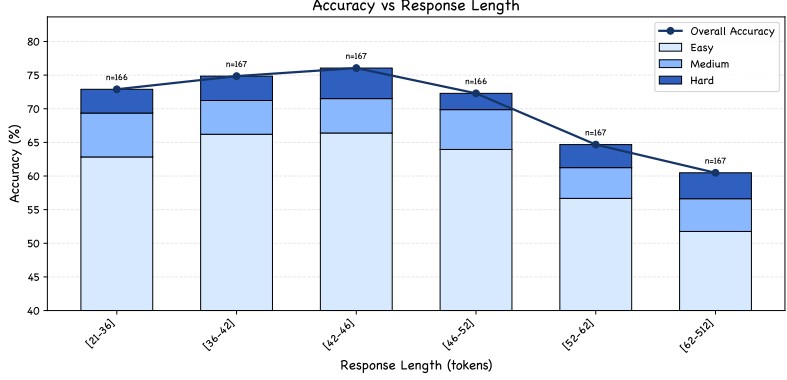

Figure 8: The trend of length and accuracy for the GRDR with model-perspective difficulty.

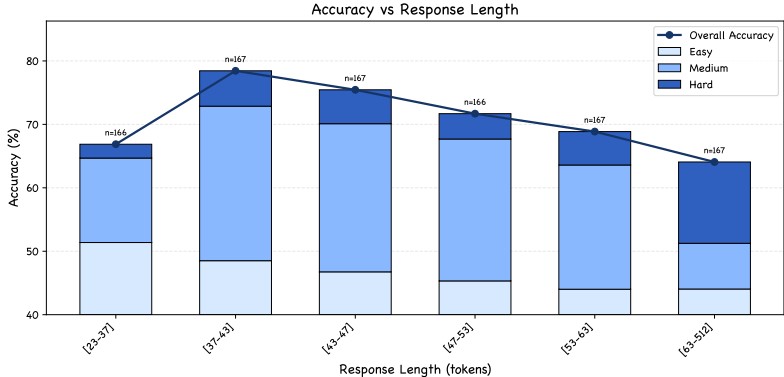

Figure 9: The trend of length and accuracy for the GA$^2$DR with human-perspective difficulty.

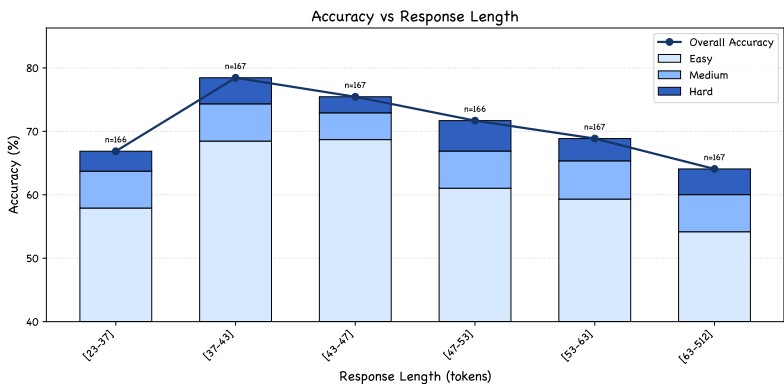

Figure 10: The trend of length and accuracy for the GA$^2$DR with model-perspective difficulty.

## A.7 TRAINING CURVES

In this section, we present the training curves of GRDR and GA$^2$DR on Qwen2.5-Omni, including grad norm, KL divergence, and reward trends in Figure 11 and Figure 12. All curves are directly extracted from the TensorBoard logs recorded during training, with a smoothing factor of 0.8 to clearly reveal their variations and overall trajectories.

Overall, both methods converge stably without exhibiting extreme values in gradients or KL divergence, and no numerical spikes are observed. This indicates that our training procedures are stable and that the proposed methods are well-behaved in practice.

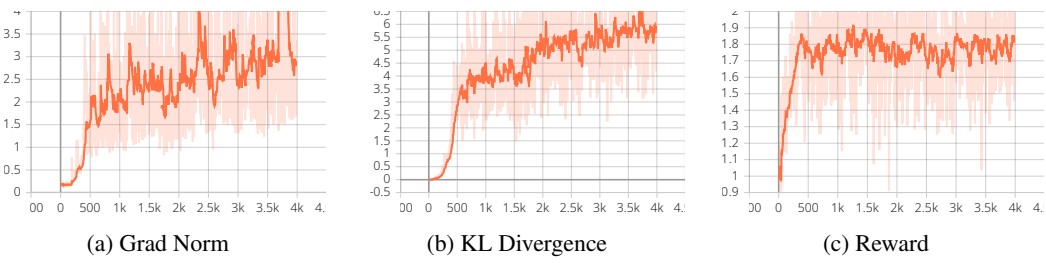

(a) Grad Norm          (b) KL Divergence          (c) Reward

Figure 11: Training curves of gradient norm, KL divergence, and reward on GRDR.

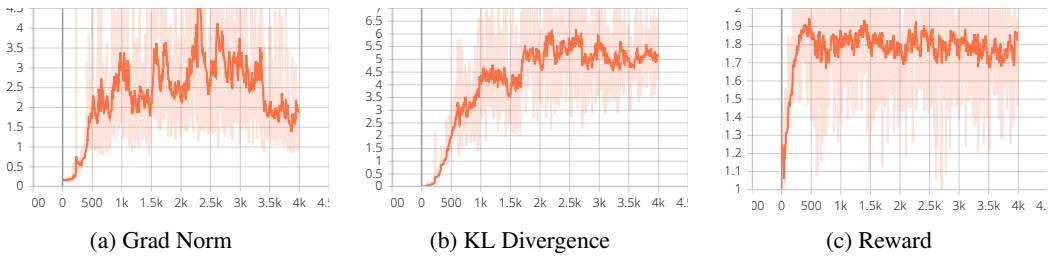

(a) Grad Norm          (b) KL Divergence          (c) Reward

Figure 12: Training curves of gradient norm, KL divergence, and reward on GA$^2$DR.

## A.8 HYPER-PARAMETERS

In this part, we provide detailed explanations of the hyperparameter settings in Table 14, including the two proposed rewards and the basic TR hyperparameter settings.

Table 14: Hyper-parameters for SFT and GRPO, including different settings under LoRA and Full.

| LoRA Fine-tuning | |
| --- | --- |
| LoRA_rank | 32 |
| LoRA_alpha | 32 |
| Torch_dtype | bfloat16 |
| Max_length | 1024 |
| Num_train_epochs | 1 |
| Per_device_train_batch_size | 8 |
| Per_device_eval_batch_size | 8 |
| Gradient_accumulation_steps | 16 |
| Learning_rate | 1.5e-6 |
| Num_generations | 8 |
| Temperature | 1.0 |
| Warmup_ratio | 0.03 |
| Beta | 0.04 |
| Epsilon | 0.2 |
| Deepspeed | zero2 |
| **Rule-base Reward** | |
| Truncation Reward $L_T$ | 120 / 400 |
| Truncation Reward $\sigma$ | -0.5 |
| GRDR and GA$^2$DR $l_{min}$ | 0.1 |
| GRDR and GA$^2$DR $k_{hard}$ | 2 |
| GRDR and GA$^2$DR $k_{easy}$ | 10 |

## A.9 CASE STUDY: QUALITATIVE ANALYSIS OF OUTPUT PARADIGMS ACROSS MODELS

In this section, we analyze output paradigms across models using 20 MMAU-test-mini questions on Dissonant Emotion Interpretation, focusing on sarcasm cause detection. The corresponding audios often involve multiple speakers and complex environments, making the task challenging and well-suited for qualitative analysis of different model outputs. In this section, in addition to the main models discussed above, we also include a Cold-Start GRPO model based on Qwen2-Audio-7B-Instruct, with detailed settings provided in the Appendix A.2.

The detailed results are shown in Table 15, which include ACC and reasoning length on the subset mentioned above. In the second part, "Based on Qwen2-Audio-7B-Instruct," we compare our three GRPO models with Audio-Reasoner. GRPO and GRPO with GRDR match Audio-Reasoner in performance while producing much shorter outputs, whereas the GRPO model after Cold-Start produces reasoning lengths close to Audio-Reasoner but performs worst. We believe this is mainly because the base model is weaker and can only imitate the surface structure of advanced model paradigms, showing that learning paradigms through Cold-Start is not effective in all cases and can even yield negative effects. In the "Based on Qwen2.5-Omni-7B" part of the table, performance differences are minor, mainly because the Qwen2.5-Omni base model already has some reasoning ability and shows good performance on complex questions. However, with our two proposed rewards, the models maintain their original performance while significantly shortening reasoning length, greatly improving reasoning efficiency.

In addition to analyzing performance and reasoning length above, we also compared the outputs of different models on these 20 questions in detail. From the results, we found that models with better performance usually follow a complete reasoning process, which includes first grounding the audio and providing a corresponding caption—equivalent to identifying the known conditions in the problem—then reasoning based on these conditions, analyzing each option step by step, and finally giving the answer. This approach helps LALMs make better use of known information together with their broad pretrained knowledge to perform reasonable reasoning while also improving readability. However, weaker models, even when using this structure, often make mistakes in the first step of extracting known information. These errors propagate through the reasoning process, leading to wrong answers and lower performance. For this part, we also provide an example of different models' outputs in Appendix A.10 as reference.

Table 15: Performance and reasoning length of different models on 20 sarcasm cause detection tasks.

| Model | ACC | Avg-Length |
|---|---|---|
| Advanced Proprietary Model | | |
| Gemini2.5-Pro-0506 | 95 | 931.8 |
| Based On Qwen2-Audio-7B-Instruct | | |
| Audio-Reasoner | 75 | 547.1 |
| GRPO | 75 | 50.7 |
| + GRDR | 70 | 38.5 |
| + Cold-Start SFT | 55 | 541.4 |
| Based On Qwen2.5-Omni-7B | | |
| GRPO | 100 | 109.5 |
| + GRDR | 95 | 56.2 |
| + GRDR ($l_{min} = 0.1$) | 90 | 94.1 |
| + GA$^2$DR | 100 | 56.9 |
| + GA$^2$DR ($l_{min} = 0.1$) | 95 | 94.2 |

Overall, we believe that a good reasoning paradigm should consist of three parts: extracting known conditions, reasoning and analysis, and giving the final answer. Among these steps, the earlier ones have a stronger impact on later results, and once errors occur, they are difficult to correct in the following steps. Therefore, we suggest that for weaker models, it is better to first fine-tune them on basic tasks related to extracting known conditions (e.g., grounding and caption) before learning reasoning paradigms. For stronger models that still produce reasoning with poor readability, Cold-Start can be used to learn the paradigm first, followed by RL training. For models that already generate reasoning with good structure, direct RL with our two proposed rewards can further improve both performance and efficiency.

## A.10 EXAMPLE ANALYSIS

In this section, we present output examples from six different models on a sarcasm cause detection task from Figure 13 to Figure 15. This question is labeled as Medium in the original MMAU-test-mini but is relabeled as Hard under our new scheme. The models include an advanced proprietary model, Gemini2.5-Pro, the prior work Audio-Reasoner, and the four main models used in our experiments. In the figure, the green text corresponds to the correct option for this example, the blue parts in each model's reasoning indicate logical and necessary steps or key points for solving the question, and the red parts indicate repetition or redundancy, representing the parts that need optimization.

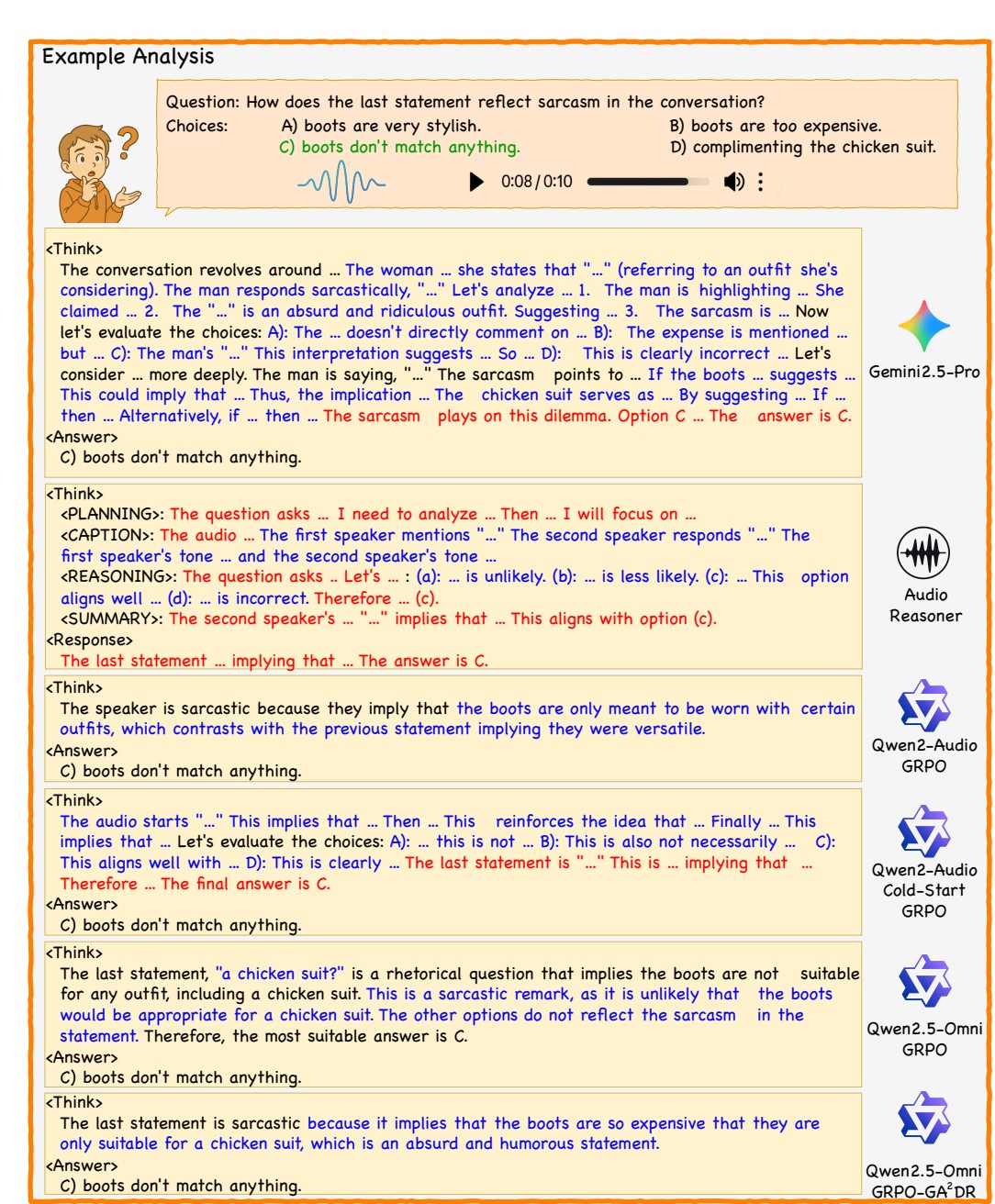

Figure 13: Output examples from six different models on a sarcasm cause detection question. The green part in the question indicates the correct option. The blue parts in the model outputs represent core content, while the red parts indicate redundancy.

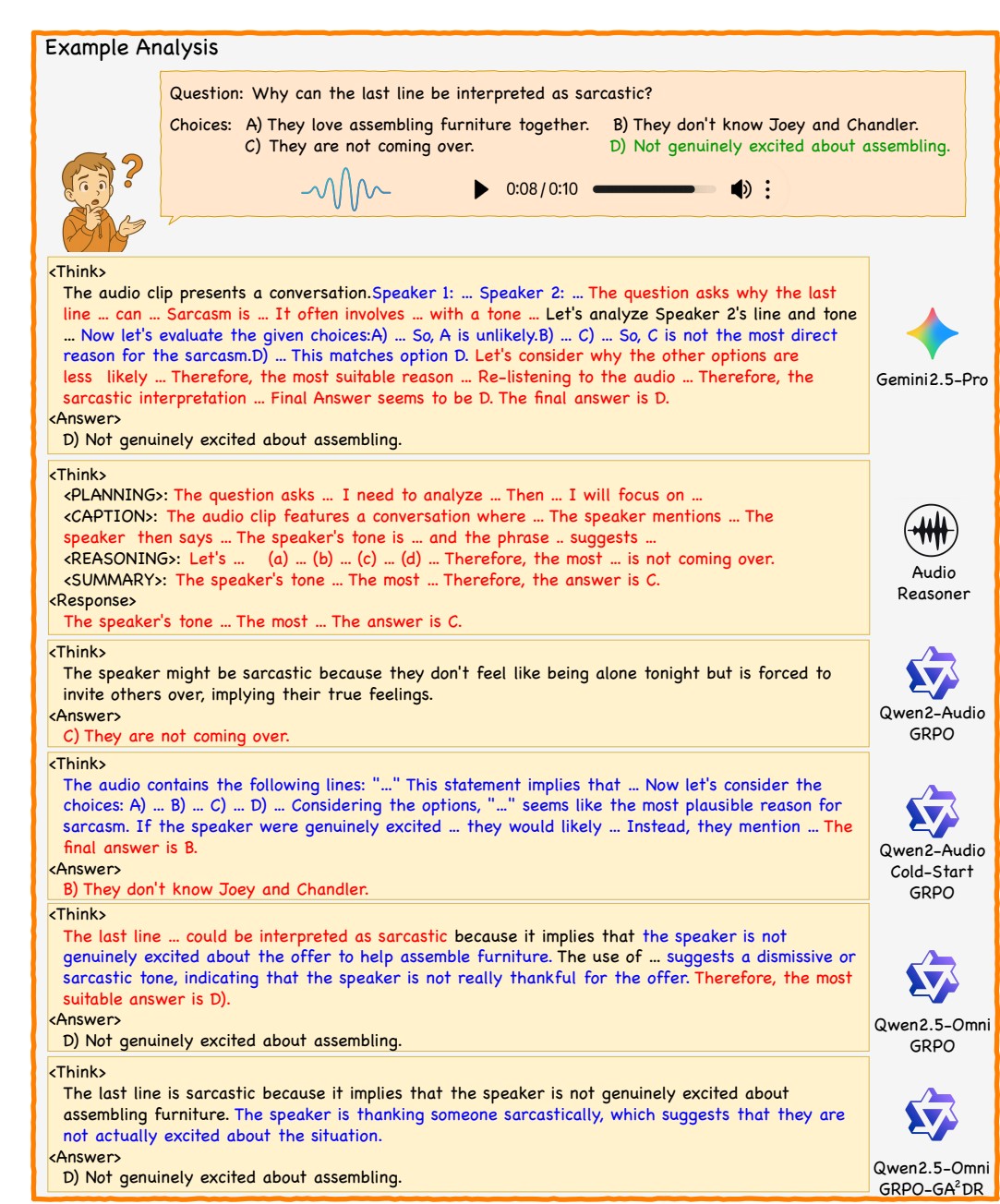

Figure 14: Output examples from six different models on a sarcasm cause detection question. The green part in the question indicates the correct option. The blue parts in the model outputs represent core content, while the red parts indicate redundancy.

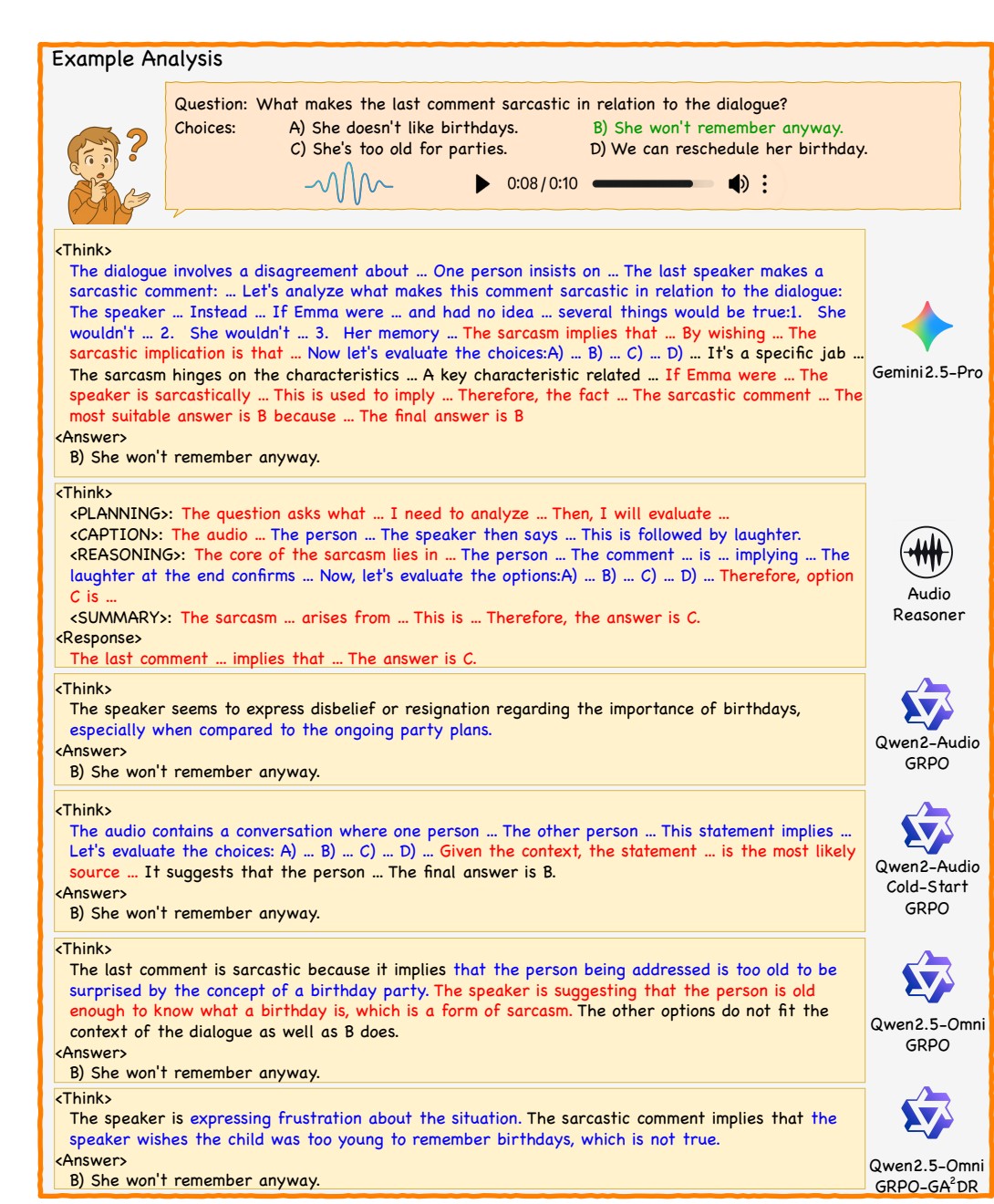

Figure 15: Output examples from six different models on a sarcasm cause detection question. The green part in the question indicates the correct option. The blue parts in the model outputs represent core content, while the red parts indicate redundancy.

