# OpenReview forum: "Think Smart, Not Hard: Difficulty Adaptive Reasoning for Large Audio Language Models"
_ICLR.cc/2026/Conference — ICLR 2026 Conference Withdrawn Submission_

### Official Review · Reviewer_gZvn · 2025-10-20

**Soundness:** 2
**Presentation:** 3
**Contribution:** 2
**Rating:** 4
**Confidence:** 4

**Summary:**

This paper presents Think Smart, Not Hard, a reinforcement-learning framework that enables difficulty-adaptive reasoning for Large Audio Language Models (LALMs). The key insight is that current LALMs, trained via SFT or GRPO, apply uniform reasoning depth regardless of question difficulty—causing redundant “overthinking” for easy tasks and shallow reasoning for complex ones.
To address this, the authors propose two difficulty-adaptive, length-based reward functions: (1) Group Ratio Difficulty Reward (GRDR), which estimates difficulty from the correctness ratio of sampled rollouts, and (2) Group Audio-Attention Difficulty Reward (GA2DR), which derives difficulty from attention-entropy over audio tokens. These rewards dynamically link reasoning length to model-perceived difficulty, encouraging concise reasoning for simple inputs and deeper reasoning for challenging ones. Experiments demonstrate consistent gains over standard GRPO and truncation-based baselines, achieving shorter reasoning traces without sacrificing accuracy.

**Strengths:**

(1) The paper identifies an important inefficiency in current LALMs by showing that they apply uniform reasoning depth across different difficulty levels. It provides a clear motivation and strong empirical support for introducing difficulty-adaptive reasoning.

(2) The proposed GRDR and GA2DR rewards are simple and effective, allowing adaptive reasoning without external supervision.

(3) Experiments on MMAU with supporting AirBench results demonstrate consistent gains over GRPO and truncation baselines.

**Weaknesses:**

(1) The paper introduces exponential difficulty–length scaling within GRPO without analyzing its impact on policy-gradient variance or monotonic-improvement conditions. In the absence of bounded-gradient or convergence guarantees, objective stability remains unverified.

(2) GA2DR uses batch-normalized attention entropy as a continuous difficulty signal (Eq. 5), but provides no calibration or correlation against human difficulty labels. Despite normalization, the signal may still conflate input noise or sequence length with true task complexity.

(3) Because GRDR derives difficulty from on-policy rollout correctness, the reward baseline drifts as training evolves, creating a feedback loop that can distort credit assignment and destabilize late-stage optimization.

(4) The Cold-Start variant shows degraded outcomes, yet the paper omits optimization diagnostics (e.g., KL, gradient-norm, reward-variance trajectories), leaving claims of adaptive-reward stability anecdotal.

(5) Reasoning length is used as a proxy for efficiency, but no inference-time, FLOP, or latency measurements are reported, so computational gains are not established beyond token-level proxies.

(6) Model-perspective difficulty labels are produced by the same model family used for training (Qwen variants among the labelers and learners), risking family-bias. No cross-family labeling or out-of-domain validation is provided to demonstrate generalization.

(7) The paper includes a single-task qualitative example highlighting redundancy vs. core reasoning, but lacks a systematic failure-mode analysis across tasks/difficulties, so it remains unclear whether shorter traces consistently reflect better reasoning rather than premature truncation.

**Questions:**

(1) How stable are the GRDR and GA2DR difficulty signals throughout training? Please provide curves of γ variance and reward distribution across epochs to verify that the adaptive scaling converges rather than oscillates or collapses.

(2) GA2DR assumes that attention entropy reflects reasoning difficulty, yet this metric could be sensitive to batch normalization, padding, or input noise. Have you examined its noise robustness or its correlation with human-annotated or dataset-defined difficulty levels on MMAU?

(3) Could the authors quantify the relationship between reasoning length and task accuracy, for instance by plotting accuracy/reward versus token length?

(4) Does the proposed difficulty-adaptive training generalize beyond audio–text reasoning to text-only or audio–visual tasks?

(5) Since the model-perspective difficulty labels are produced by the same family of backbones (Qwen2-Audio, Qwen2.5-Omni, Kimi-Audio, Gemini2.5-Pro) used for training, could there be self-consistency or family bias? Have you tested cross-model validation using an unseen architecture to confirm that the learned adaptivity generalizes?

(6) What is the per-batch computational overhead of calculating attention-entropy–based difficulty, and can it scale to larger multimodal corpora or longer sequences without becoming a bottleneck?

(7) The method emphasizes shorter reasoning chains, but is there evidence that reasoning quality remains unchanged? Have you conducted any human or LLM-as-a-judge evaluations of coherence, factual correctness, or interpretability to ensure that brevity does not harm reasoning fidelity?

(8) The Cold-Start SFT→GRPO variant exhibits degraded performance, but no diagnostics are shown. Could the authors provide gradient-norm, reward-variance, or KL divergence curves to clarify why Cold-Start fails to stabilize and whether this reflects optimization instability or data mismatch?

---

> ### Author Response · Authors · 2025-11-17
>
> # Response To Reviewer gZvn
>
> Thank you for your valuable comments! In response to the mentioned weaknesses and questions, our replies are as follows. Please also refer to our general comment, which summarizes the main updates in this version and addresses common problems shared by multiple reviewers.
>
> ## Weakness 1, 4 & Question 1, 8: About the training curves
>
> In the new version, we include complete training curves for $GRDR$ and $GA^2DR$ (grad norm, KL divergence, and reward trends) in Appendix A.7, Figures 11–12. The results show smooth convergence without large oscillations, confirming the stability of our methods.
>
> The Cold-Start experiment explores reasoning paradigms. Its SFT data are distilled from Gemini2.5-Pro and optimized with Qwen3, while the RL stage uses standard GRPO without our rewards. Thus, the weaker results in some samples are unrelated to our methods.
>
> However, to address the concern you raised, we provide the complete evaluation results of our Cold-Start + GRPO experiment below. The results show that the training performs well.
>
> | Models (On MMAU-Test-Mini)                           | Sound     | Music     | Speech    | Avg       |
> | ---------------------------------------------------- | --------- | --------- | --------- | --------- |
> | R1-AQA (Qwen2-Audio-7B-Instruct + Full-GRPO)         | 68.77     | 64.37     | 63.66     | 65.60     |
> | Ours (Qwen2-Audio-7B-Instruct + Cold-Start Full-SFT) | 63.06     | 56.59     | 57.66     | 59.10     |
> | $\quad$+ Full-GRPO On AVQA                                  | 67.27     | 66.17     | 59.46     | 64.30     |
> | $\quad$+ Full-GRPO On FS                                    | **70.57** | **65.57** | **66.07** | **67.40** |
>
> ## Weakness 3: About the reward drift as training evolves
>
> Regarding the concern that $GRDR$ may drift in difficulty estimation and form a feedback loop, we believe this refers to potential reward hacking, where reasoning becomes overly short. This issue is minor on simple benchmarks but appears on more complex ones like MMAR (Section 4, Table 5; Appendix A.4, Table 12). Introducing the threshold $l_{min}$ effectively mitigates this by preventing excessive shortening and reducing the likelihood of hacking.
>
> ## Weakness 5: About the efficiency during training and inferring
>
> All our methods operate only on the training side, while inference uses the same base models without extra modules. Thus, inference efficiency can be fairly compared by measuring output token length.
>
> During training, both methods reuse existing variables without adding complex computation. Although $GA^2DR$ involves attention calculations, their cost is negligible compared to rollout time, so neither method becomes a bottleneck, even for long sequences.
>
> ## Weakness 6 & Question 5: About the model-perspective difficulty
>
> In our paper, we use model-perspective difficulty in two ways, and we apologize for the confusion. For evaluation, we obtain the model-perspective difficulty on MMAU from the accuracy proportions of four models. These labels are used only for evaluation and are not involved in training.
>
> As for the four models, Qwen2-Audio and Qwen2.5-Omni are developed by the Qwen team, but they differ in architecture and pretraining. The other two models come from different companies, so they are not from the same model family.
>
> ## Weakness 7 & Question 7: About more qualitative example
>
> In the previous version of the paper, we only provided one qualitative example. In this version, we add two additional examples to further illustrate our findings. Moreover, the anonymous link included in the paper contains the complete reasoning outputs of all models for all samples within the entire task as a reference.
>
> ## Question 3: About the relationship between response length and accuracy
>
> In Appendix A.6, we present four plots of response length versus accuracy. $GRDR$ performs better at shorter lengths, while $GA^2DR$ excels in the mid-length range. Under human annotations, correct answers show no clear difficulty pattern, but under the model-perspective view, the distribution is more balanced, confirming the effectiveness of using model-based difficulty in training.
>
> ## Question 4: About the generalization of our methods on other modality tasks
>
> Although our work focuses on audio, both methods are theoretically transferable. $GRDR$, based on rollout accuracy, can be applied to any GRPO-style methods, while $GA^2DR$ can adapt to different modalities—focusing on text for language models or combining attention across audio and vision for multimodal ones. Thus, our approach generalizes well beyond audio tasks.
>
> Thank you again for your thoughtful comments, and we hope our answers have adequately addressed your doubts.

---

> > ### Author Response · Authors · 2025-11-24
> >
> > Dear Reviewer gZvn.
> >
> > $\quad$We would greatly appreciate it if you could review and respond to our rebuttal. We have added the training curves, complete cold-start results, and several clarifications, and we would greatly appreciate knowing if our response satisfactorily addressed your concerns. If anything remains unclear, please let us know. We look forward to your feedback throughout the Rebuttal Period.
> >
> > Thank you!

---

### Official Review · Reviewer_2MYJ · 2025-10-28

**Soundness:** 2
**Presentation:** 2
**Contribution:** 2
**Rating:** 2
**Confidence:** 3

**Summary:**

This paper tries to address the "one-size-fits-all" issue in LALMs reasoning, analyzing performance differences between SFT/GRPO and the impact of explicit/implicit prompts. It proposes two adaptive reward functions: GRDR (discrete difficulty) and GA²DR (continuous difficulty). Experiments on the MMAU benchmark verify that the method improves performance while shortening reasoning length, and also refines an ideal reasoning paradigm and model training recommendations, providing a solution for LALMs reasoning optimization.

**Strengths:**

- It identifies the pain point of difficulty adaptation in LALMs reasoning, fills gaps in SFT/GRPO comparison and prompt effect analysis.
- The method aligns with audio modal characteristics; the two difficulty definitions cover different scenarios, featuring flexible design.
- Experiments use multiple baselines and multi-dimensional verification, and introduce model-perspective difficulty labeling, ensuring reliable results.

**Weaknesses:**

- GRDR uses a fixed group size G=8 without verifying other values, and GA²DR lacks basis for layer selection, leading to insufficient robustness in difficulty definition.
- The hyperparameter values of the reward function have no basis and lack sensitivity analysis, affecting reproducibility.
- There are inconsistencies in formula and figure formatting. such as no periods or commas are added at the end of formulas. Figure 1 lacks a legend, which makes it impossible to identify the meaning of different curves, colors, or markers in the figure.

**Questions:**

1. (Line 192) What is the basis for choosing G=8 in GRDR? How will performance change if G is set to 4 or 16?
2. (Line 211) Why does GA²DR use the last layer to calculate attention entropy? Will switching to an intermediate layer cause significant differences?
3. (Line 246) What is the basis for setting $k_{easy}$ and $k_{hard}$ in the reward function? How will performance change after adjustment?

---

> ### Author Response · Authors · 2025-11-17
>
> # Response To Reviewer 2MYJ
>
> Thank you for your valuable comments! In response to the mentioned weaknesses and questions, our replies are as follows. Please also refer to our general comment, which summarizes the main updates in this version and addresses common problems shared by multiple reviewers.
>
> ## Weakness 1, 2 & Question 1, 2, 3: About the G in GRDR and several hyper-parameters
>
> These issues are common concerns raised by multiple reviewers, and we have addressed it collectively in the general response above. The concerns in this part are all addressed in Common Problem 3. Thank you.
>
> ## Weakness 3: About some trouble in formula and figure formatting
>
> We have reviewed the entire paper and corrected several formulas and figures, including the previously missing legend in Figure 2. Thank you.
>
> We are grateful for your careful review and hope that our responses have provided the clarification you needed.

---

> > ### Author Response · Authors · 2025-11-24
> >
> > Dear Reviewer 2MYJ.
> >
> > $\quad$We would sincerely appreciate your review of our rebuttal. We have improved the clarity of the paper, including revisions to Figure 2, and added further details to address the corresponding concerns. We would greatly appreciate knowing if our response satisfactorily addressed your concerns. If you have any additional inquiries, we are happy to provide further information. We look forward to your feedback throughout the Rebuttal Period.
> >
> > Thank you!

---

### Official Review · Reviewer_vJnZ · 2025-10-28

**Soundness:** 2
**Presentation:** 3
**Contribution:** 2
**Rating:** 4
**Confidence:** 5

**Summary:**

The paper studies how Large Audio‑Language Models (LALMs) should modulate how much they reason, rather than simply deciding whether to reason. Building on analyses that compare SFT vs. GRPO and explicit vs. implicit CoT prompting, the authors propose difficulty‑adaptive reward shaping that links reasoning length to problem difficulty. They introduce two difficulty estimators computed from the model’s perspective: GRDR (Group Ratio Difficulty Reward), which labels a question as easy/medium/hard based on the share of correct rollouts in a group and GA$^2$DR (Group Audio‑Attention Difficulty Reward), which maps the entropy of last‑token attention over audio tokens to a continuous difficulty $\gamma \in [0,1]$. A negative‑exponential length reward promotes short, concise CoT on easy items and longer, exploratory CoT on hard ones. Experiments on MMAU‑test‑mini with Qwen2‑Audio‑7B and Qwen2.5‑Omni‑7B (LoRA + GRPO, explicit CoT prompt) show modest average accuracy gains over GRPO+TR and notably bigger gains on hard subsets.

**Strengths:**

- Paper is clear and easily understandable
- Meaningful improvements on hard items with small overall gain
- Good ablation on explicit CoT helps on hard items.

**Weaknesses:**

- The paper has limited evaluation scope. Results focus on MMAU‑test‑mini with single‑run ACC. There are no standard errors or full‑benchmark experiments.
- The paper doesn't report compute/training cost: GRDR needs grouped rollouts; GA$^2$DR requires attention extraction. The paper does not quantify added RL cost vs. GRPO/TR.
- The hyperparameters are not ablated. For instance, among slopes ($k_{easy}$ and $k_{hard}$), Group sizer $G$ and $l_{min}$; only $l_{min}$ is ablated.
- GA$^2$DR hinges on last‑token attention over audio tokens, however, the scheme is not tested across across different layers/heads. Why the last year only?
-  Fig. 2 shows log‑length trends but not absolute token reductions on the full test. The log trend hides effective sizes and makes it hard to tell how much shorter the reasoning actually got.
- The paper measures all lengths with the Qwen2-Audio tokenizer, even when the model being trained/evaluated is Qwen2.5-Omni. This might lead to discrepancies if two models are using different tokenizers.

**Questions:**

- What is the training overhead of GRDR/GA$^2$DR relative to GRPO and TR (same batch size, same G)?
- For GA$^2$DR, does using different layers/heads (or pre‑softmax attention) change the difficulty estimates and outcomes?
- Do results hold on the full MMAU or other audio‑reasoning datasets?

---

> ### Author Response · Authors · 2025-11-17
>
> # Response To Reviewer vJnZ
> Thank you for your valuable comments! In response to the mentioned weaknesses and questions, our replies are as follows. Please also refer to our general comment, which summarizes the main updates in this version and addresses common problems shared by multiple reviewers.
>
> ## Weakness 1 & Question 3: About more evaluations
>
> These issues are common concerns raised by multiple reviewers, and we have addressed it collectively in the general response above. Weakness 1 and Question 3 are both answered in Common Problem 2. Thank you.
>
> ## Weakness 2 & Question 1: About the training cost
>
> These issues are common concerns raised by multiple reviewers, and we have addressed it collectively in the general response above. Weakness 2 and Question 1 are both addressed in Common Problem 4. Thank you.
>
> ## Weakness 3, 4 & Question 2: About hyper-parameters and attention settings
>
> These issues are common concerns raised by multiple reviewers, and we have addressed it collectively in the general response above. Concerns in this part all correspond to Common Problem 3. Thank you.
>
> ## Weakness 5 & 6: About some evaluation settings
>
> In this version, we add Figure 3, which shows only the GRPO model and our two methods based on Qwen2.5-Omni, with lengths presented directly to clearly show variation across difficulty levels. Detailed reasoning lengths for all models on MMAU-Test-Mini are provided in Appendix A.5, Table 13.
>
> For length evaluations, we use the Qwen2-Audio tokenizer, identical to Qwen2.5-Omni’s, ensuring consistency. Kimi-Audio, initialized from Qwen2.5, also has no length discrepancy. Gemini2.5-Pro is closed-source, so its tokenizer is inaccessible, but since its reasoning length is several times longer, minor deviations do not affect the overall trend.
>
> Thank you once more for taking the time to review our paper. We hope our clarifications have answered your concerns.

---

> ### Comment · Reviewer_vJnZ · 2025-11-21
>
> Thank you for responding to my queries. I will change my rating accordingly.

---

> > ### Author Response · Authors · 2025-11-22
> >
> > Dear Reviewer vJnZ:
> >
> > $\quad$Thank you very much for reading our rebuttal. We truly appreciate your willingness to reconsider your evaluation. If you have any further questions, we would be delighted to provide any additional information. Thank you.

---

### Official Review · Reviewer_8amV · 2025-10-29

**Soundness:** 3
**Presentation:** 3
**Contribution:** 3
**Rating:** 6
**Confidence:** 4

**Summary:**

This paper tackles the inefficiency of LALMs using a uniform reasoning depth for all tasks. The authors propose a "difficulty-adaptive" method, using a new RL reward function that links reasoning length to the model's perceived difficulty. This encourages concise reasoning for simple tasks and in-depth reasoning for complex ones, ultimately improving performance while significantly shortening the reasoning length.

**Strengths:**

1. It addresses the critical and practical problem of high computational cost and inefficiency in CoT reasoning, achieving the important goal of "better performance with less computation."
2. The core idea of using "model-perspective difficulty" (instead of static human labels) is intelligent. The proposed metrics, like $GA^{2}DR$ (based on audio attention entropy), are novel and well-justified.
3. The experimental results are strong. The method not only improves accuracy on hard problems but also significantly reduces the average reasoning length, effectively validating the "think smart" concept.

**Weaknesses:**

1. The experimental validation relies almost entirely on the MMAU-test-mini benchmark. While the analysis on this benchmark is deep (e.g., broken down by difficulty), this raises questions about generalizability.
2. The proposed difficulty metrics (GRDR and $GA^{2}DR$) appear computationally expensive during training. GRDR requires $G=8$ full rollouts per sample just to determine its difficulty level1. $GA^{2}DR$ requires extra computation for attention entropy and relies on batch normalization 2222. This overhead could make the already complex RL training process significantly slower and more costly, a trade-off that is not discussed in detail.

**Questions:**

1. Regarding the GRDR metric , how sensitive are the final performance and efficiency gains to the specific thresholds you chose (e.g., 3 and 6)? Would relaxing the definition of "easy" from 6 correct rollouts to 5, for example, significantly alter the results?
2. In Section 4.3, you analyze and identify an ideal reasoning paradigm: extracting conditions (captioning), followed by analysis, and then the answer. However, your reward function (Equation 6) primarily optimizes for length and correctness. How does this reward mechanism ensure that as the model shortens its reasoning, it preserves this critical logical structure rather than simply deleting random thought steps?

---

> ### Author Response · Authors · 2025-11-17
>
> # Response To Reviewer 8amV
> Thank you for your valuable comments! In response to the mentioned weaknesses and questions, our replies are as follows. Please also refer to our general comment, which summarizes the main updates in this version and addresses common problems shared by multiple reviewers.
>
> ## Weakness 1: About the generalizability on other benchmarks
>
> These issues are common concerns raised by multiple reviewers, and we have addressed it collectively in the general response above. For the response to this part, please refer to Common Problem 2. Thank you.
>
> ## Weakness 2 & Question 1: About definition and computation cost of $GRDR$&$GA^2DR$
>
> These issues are common concerns raised by multiple reviewers, and we have addressed it collectively in the general response above. Weakness 2 is answered in the first point of Common Problem 4, and Question 1 is answered in Common Problem 1. Thank you.
>
> ## Question 2: About the reasoning paradigm
>
> In Section 4.3 of the previous version, we discussed the reasoning paradigm. This paradigm is only an analysis and exploration of the outputs from different models, including our own, showing that our method does not shorten reasoning randomly and that key steps are preserved. However, we do not use this paradigm as a reward constraint during training. In future work, we plan to explore this direction further, designing rewards based on evaluating rollout paradigms with LLM APIs. More details can be found in the newly added Section 6, Limitations and Future Work.
>
> We sincerely appreciate your review and hope that our replies have addressed your questions.

---

> > ### Author Response · Authors · 2025-11-24
> >
> > Dear Reviewer 8amV.
> >
> > $\quad$Based on your insightful suggestions, we have added new benchmarks and provided detailed clarifications on several key points. We would greatly appreciate knowing if our response satisfactorily addressed your concerns, and if any issues still remain, please feel free to let us know. We look forward to your feedback throughout the Rebuttal Period.
> >
> > Thank you!

---

### Official Review · Reviewer_73Us · 2025-10-31

**Soundness:** 3
**Presentation:** 3
**Contribution:** 2
**Rating:** 4
**Confidence:** 2

**Summary:**

This paper proposes a difficulty-adaptive reasoning framework for LALMs, introducing two self-perspective difficulty estimators and length-negative-exponential rewards (GRDR and GA²DR). The goal is to encourage concise reasoning for easy questions and deeper reasoning for hard ones, thereby improving efficiency without sacrificing accuracy. The authors conduct experiments on the MMAU benchmark, demonstrating that their approach reduces reasoning length while maintaining strong performance.

**Strengths:**

The motivation is compelling, addressing a critical limitation of current LALMs—lack of adaptive reasoning. The proposed rewards (GRDR and GA²DR) show practical benefits, significantly reducing reasoning length on the MMAU-test-mini benchmark while preserving accuracy(From Table 4). The framework’s ability to dynamically adjust reasoning depth based on question difficulty is a promising direction for improving LALM efficiency.

**Weaknesses:**

1、Contribution 1 from authors only reports aggregate accuracy on MMAU-Test-Mini split by difficulty.  The authors should perform finer-grained statistical analyses to validate the claim that “models under-/over-reason on different difficulties”.
2、Contribution 2 from authors introduces GRDR and GA²DR, but their relationship is unclear.  GA²DR appears to be a stand-alone criterion; its connection with GRDR is never formally discussed.  Besides, GRDR itself seems questionable: Table 2 shows that models perform best on medium-difficulty questions, while easy and hard accuracies are almost equal.  Under GRDR’s definition, should the original three difficulty buckets be re-labeled?  If so, how does the new labeling relate to the original one, and why is it more reasonable?
3、How are the “groups” in GRDR constructed?
4、What is the theoretical or empirical link between GRDR and GA²DR?  Table 3 treats them as independent ablations.  If they are orthogonal, why must both be proposed?  Moreover, the performance gaps in Table 3 are within one standard deviation, so improvements may simply come from random-seed variation rather than the rewards themselves.
5、Experiments are restricted to MMAU-test-mini. No results on other audio-reasoning benchmarks or cross-domain transfer are provided, limiting generalizability.
6、Line 361 claims “Specifically, it achieves short reasoning for simple questions and long reasoning for difficult ones, while overall significantly reducing reasoning length.”  But Figure 2 only shows that GRDR/GA²DR shorten the average length; it gives no evidence that the model produces longer chains for hard questions than for easy ones.

**Questions:**

See Weaknesses

---

> ### Author Response · Authors · 2025-11-17
>
> # Response To Reviewer 73Us
> Thank you for your valuable comments! In response to the mentioned weaknesses, our replies are as follows. Please also refer to our general comment, which summarizes the main updates in this version and addresses common problems shared by multiple reviewers.
>
> ## Weakness 1 & 6: About fine-grained statistical analysis on under-reasoning / over-reasoning
>
> We report average accuracy across difficulty levels in Table 2. SFT without reasoning performs well on easy questions but poorly on hard ones, while GRPO with reasoning performs better on hard questions but worse on easy ones. This means that, for GRPO, easy questions are over-reasoned, and for SFT, hard questions require further reasoning. Figures 2 and 3 further confirm this, with our methods achieving higher accuracy using shorter CoTs. We also provide the full length statistics in Appendix A.5, Table 13 for reference.
>
> ## Weakness 2: About the difficulty labels
>
> The difficulty labels in Table 2 are annotated based on the accuracy proportions of four models on the MMAU evaluation set. These labels are used only for evaluation and are unrelated to the difficulty used during training.
>
> ## Weakness 3, 4 & 5: About the definition of group, $GRDR$&$GA^2DR$ and more evaluations on other benchmarks
>
> These issues are common concerns raised by multiple reviewers, and we have addressed it collectively in the general response above. Weakness 3 is addressed in the first point of Common Problem 3. Weakness 4 is addressed in Common Problem 1, and Weakness 5 is addressed in Common Problem 2.
>
> Thank you again for your review, and we hope our responses have resolved your concerns.

---

> > ### Author Response · Authors · 2025-11-24
> >
> > Dear Reviewer 73Us.
> >
> > $\quad$Following your helpful suggestions, we have added detailed explanations and results on two additional benchmarks. We would greatly appreciate knowing if our response satisfactorily addressed your concerns. If any points remain unclear, please feel free to let us know. We are willing to provide any additional information throughout the entire Rebuttal Period.
> >
> > Thank you!

---

### Author Response · Authors · 2025-11-17

# General Response

## Updates in this edition of the paper

### Added more experiments:
1. We add the results of our methods on two new benchmarks, presented in Table 4 and Table 5 of Section 4, and Table 11 and Table 12 in Appendix A.4. (suggested by Reviewer 73Us, 8amV and vJnZ)
2. We include additional ablation studies and experiments with different k-value settings, reported in Section 4.3. (suggested by Reviewer vJnZ and 2MYJ)

### Improved the clarity:

1. In Appendix A.1, we add Figure 4 to illustrate the task definition.
2. We refine several unclear descriptions and update some ambiguous formulas and figures. (suggested by Reviewer 2MYJ)
3. In Table 13 of Appendix A.5, we provide the detailed reasoning lengths of all models on MMAU-Test-Mini. (suggested by Reviewer 73Us)
4. In Figures 7 to 10 of Appendix A.6, we add the variation plots of reasoning length and accuracy for $GRDR$ and $GA^2DR$ on MMAU-Test-Mini. (suggested by Reviewer gZvn)
5. In Figures 11 and 12 of Appendix A.7, we present the training curves of $GRDR$ and $GA^2DR$. (suggested by Reviewer gZvn)
6. In Appendix A.10, we add two additional example analyses, Figure 14 and Figure 15. (suggested by Reviewer gZvn)

---

### Author Response · Authors · 2025-11-17

# General Response
## Commom Problem 1: About the definitions and relationship of $GRDR$&$GA^2DR$ (To Reviewer 73Us and 8amV)
In training, our difficulty estimation comes from two perspectives. $GRDR$ is outcome-oriented, where difficulty is determined by the correctness ratio of rollout responses—a higher ratio indicates an easier question. Specifically, it divides the eight rollout responses (corresponding to nine possible cases) evenly into three difficulty levels. $GA^2DR$ is process-oriented, where difficulty is measured using audio attention entropy—a more dispersed attention distribution corresponds to higher entropy and greater difficulty. Experimental results show that both methods achieve strong performance.
## Common Problem 2: About more evaluations on other benchmarks (To Reviewer 73Us, 8amV and vJnZ)
For ease of reading, we present the results on the two newly added benchmarks below.
We update our paper with two new benchmarks: MMAU-v0515 and MMAR. MMAU-v0515 refines several Q&A pairs and audio samples, while MMAR focuses on complex, cross-lingual reasoning tasks with more diverse and challenging data. Across both benchmarks, our methods maintain leading performance, further demonstrating their strong generalization ability.
| Models (On MMAU-Test-Mini-v0515) | Sound | Music | Speech | Avg       |
| -------------------------------- | ----- | ----- | ------ | --------- |
| Qwen2.5-Omni-7B                  | 74.17 | 65.26 | 61.56  | 67.00     |
| GRPO                             | 84.08 | 69.46 | 74.17  | 76.30     |
| $\quad$+ TR                             | 83.78 | 70.65 | 74.47  | 76.30     |
| $\quad$+ GRDR                           | 83.48 | 70.35 | 75.97  | 76.60     |
| $\qquad$+ $l_{min}=0.1$                  | 83.18 | 70.95 | 76.27  | 76.80     |
| $\quad$+ $GA^2DR$                       | 83.18 | 71.55 | 75.67  | 76.80     |
| $\qquad$+ $l_{min}=0.1$                  | 81.08 | 72.15 | 77.77  | **77.00** |

| Models (On MMAR) | Sound | Music | Speech | Avg       |
| ---------------- | ----- | ----- | ------ | --------- |
| Qwen2.5-Omni-7B  | 58.79 | 40.78 | 59.86  | 56.70     |
| GRPO             | 60.00 | 48.05 | 62.24  | 59.90     |
| $\quad$+ TR             | 64.84 | 49.51 | 63.94  | 61.90     |
| $\quad$+ GRDR           | 61.21 | 51.94 | 65.30  | 61.20     |
| $\qquad$+ $l_{min}=0.1$  | 63.63 | 52.91 | 65.98  | 63.00     |
| $\quad$+ $GA^2DR$       | 64.84 | 54.85 | 65.30  | 62.90     |
| $\qquad$+ $l_{min}=0.1$  | 64.84 | 53.39 | 63.60  | **63.00** |

---

### Author Response · Authors · 2025-11-17

# General Response
## Common Problem 3: About several hyps-parameters (G, k, $l_{min}$, Attention-layers / heads) (To Reviewer 73Us and 2MYJ)

1. **About the G in $GRDR$**: In $GRDR$, representing the number of responses rolled out for a single example. Following prior works (R1-AQA and Audio-Thinker), we set G = 8.

2. **About the k in the length-based reward**: We perform an ablation study by removing the dynamic $k(\gamma)$ and using fixed k values to validate our method. Different k settings are tested across three benchmarks, with results reported in Section 4.3 and partial data shown below. Here "*" represent the model-perspective difficulty label. From the data, we can see that while the length-based reward is less effective than GRDR, it still outperforms direct GRPO and TR. Different k-value settings show minor overall differences but fluctuate notably across difficulty levels, indicating that using a fixed k for all problems causes imbalance. These results further demonstrate the effectiveness of our method.

   | Models (On MMAU-Test-Mini) | Easy  | Easy* | Medium | Medium* | Hard  | Hard* | Avg          |
   | -------------------------- | ----- | ----- | ------ | ------- | ----- | ----- | ------------ |
   | GRPO                       | 59.69 | 93.92 | 78.23  | 59.81   | 60.77 | 27.41 | 69.40 (-0.8) |
   | $\quad$+ TR                       | 58.14 | 93.73 | 78.43  | 57.47   | 57.75 | 25.86 | 68.40 (-1.8) |
   | $\quad$+GRDR (ours)               | 60.07 | 93.16 | 80.00  | 58.87   | 59.91 | 32.81 | 70.20        |
   | $\quad$+ Length-based Rewardf (k=2)                        | 59.69 | 94.49 | 77.84  | 58.41   | 61.63 | 27.41 | 69.40 (-0.8) |
   | $\quad$+ Length-based Rewardf (k=6)                        | 56.20 | 93.92 | 79.21  | 57.00   | 61.63 | 28.95 | 69.20 (-1.0) |
   | $\quad$+ Length-based Rewardf (k=10)                       | 58.91 | 93.92 | 79.60  | 56.07   | 59.48 | 31.27 | 69.60 (-0.6) |

3. **About the selection of the attention layer and heads**: In our preliminary analysis, we examined the entropy values across different layers and found that the differences were minimal. Considering that deeper layers contain richer semantic information and better reflect the model’s final problem-solving state, we therefore select the last layer. For attention heads, since different heads contribute differently and using a single head introduces uncertainty, we take the average across all heads within the same layer.

## Common Problem 4: About the computational cost of our methods (To Reviewer 8amV and vJnZ)

With the same batch size and G settings, direct GRPO, GRPO with TR, and GRPO with our rewards show no notable difference in training time or memory usage. Our methods add no extra computation since they reuse rollout accuracies and attention values from the forward pass. All methods train at about 0.0125 steps per second (≈80 seconds per step), and full training on FS takes roughly 3.6 days on three A100-40G GPUs (two for forward passes, one for vLLM rollout). In addition, when handling long sequences, the time cost of generating rollout responses is far greater than that of our difficulty computation, so our methods do not become a bottleneck.

---

### Note · Authors · 2026-01-04

I have read and agree with the venue's withdrawal policy on behalf of myself and my co-authors.